# GrabS: Generative Embodied Agent for 3D Object Segmentation without Scene Supervision

**Zihui Zhang** [1,2]    **Yafei Yang** [1,2]    **Hongtao Wen** [1,2]    **Bo Yang** [1,2]*

[1] Shenzhen Research Institute, The Hong Kong Polytechnic University
[2] vLAR Group, The Hong Kong Polytechnic University
`zihui.zhang@connect.polyu.hk`    `bo.yang@polyu.edu.hk`

## Abstract

We study the hard problem of 3D object segmentation in complex point clouds without requiring human labels of 3D scenes for supervision. By relying on the similarity of pretrained 2D features or external signals such as motion to group 3D points as objects, existing unsupervised methods are usually limited to identifying simple objects like cars or their segmented objects are often inferior due to the lack of objectness in pretrained features. In this paper, we propose a new two-stage pipeline called **GrabS**. The core concept of our method is to learn generative and discriminative object-centric priors as a foundation from object datasets in the first stage, and then design an embodied agent to learn to discover multiple objects by querying against the pretrained generative priors in the second stage. We extensively evaluate our method on two real-world datasets and a newly created synthetic dataset, demonstrating remarkable segmentation performance, clearly surpassing all existing unsupervised methods.

## 1 Introduction

The emerging applications in autonomous navigation, embodied AI, and mixed reality necessitate precise semantic 3D scene understanding. Particularly, the ability to identify complex objects in 3D point clouds is crucial for machines to reason about and interact with real-world environments. Existing works to tackle 3D object segmentation mainly rely on dense or sparse human labels for 3D supervision (Wang et al., 2018; Schult et al., 2023), or large-scale image/language annotations for 2D-3D supervision (Takmaz et al., 2023; Yin et al., 2024). Although they have achieved excellent segmentation results, the required large-scale annotations are laborious to collect, making them unappealing and less generic in real applications.

To overcome this limitation, a few unsupervised methods aim to group 3D points as objects by either relying on heuristics such as distributions of point normals/colors/motions (Zhang et al., 2023; 2024; Song & Yang, 2022; 2024), or the similarity of self-supervised pretrained point features commonly reprojected from 2D images (Rozenberszki et al., 2024; Shi et al., 2024). Despite obtaining encouraging results, they can usually identify simple objects like cars in self-driving scenarios, or their segmented objects are often inferior in quality due to the lack of objectness of pretrained features.

In this paper, we aim to design a generic pipeline that can precisely identify complex objects in 3D point clouds without requiring any costly human labels of 3D scenes for supervision. However, this is extremely challenging as it involves two fundamental questions: 1) what are objects (*i.e.*, *object priors*)? and 2) how to effectively estimate multiple these objects in complex scenes? In fact, in real 3D world, this is even harder, because different objects of the same category (*e.g.*, *chairs*) may exhibit distinctive morphologies due to severe occlusions, different orientations located, and sensory noises. This means that: 1) the yet to be defined or learned object priors should be both discriminative, robust and continuous in latent space, and 2) the yet to be designed estimation strategy should take into account possible missed detections during object exploration.

With this motivation, we introduce a new learning framework comprising two natural stages: 1) 3D object prior learning, followed by 2) object estimation of 3D scenes without needing human labels for supervision. As illustrated in the left block of Figure 1, in the first stage, we aim to train

---

*Corresponding Author

Figure 1: An illustration of the overall framework.

an **object-centric network** to learn both discriminative and robust object priors from single object shapes such as ShapeNet (Chang et al., 2015). In the second stage, as shown in the right block of Figure 1, we introduce a **multi-object estimation network** to infer multiple objects in an input point cloud, just by using learned object priors in the first stage without needing human labels to train.

For the **object-centric network**, to learn desired object priors against potential occlusions, noises, and chaotic object orientations, we choose to learn an object-centric generative model. In particular, given an object point cloud, the network aims to estimate a conditional latent distribution via existing techniques such as Variational Autoencoder (VAE) (Kingma & Welling, 2014) and diffusion model (Ho et al., 2020). The latent code is expected to be unique for a specific viewing angle, and the network could regress the object orientation with regard to a canonical pose. In this way, the learned generative object priors could be robust for occlusions or noises, whereas the orientation estimation ability would allow the learned priors to be discriminative for different object orientations.

Regarding the **multi-object estimation network**, we aim to discover individual objects as many as possible on scene-level point clouds, but only relying on our pretrained generative object-centric network. Our insight is that, given a subvolume of points cropped from the input scene point cloud, if it happens to include a single valid object, its latent priors should be able to recover/generate a plausible object shape and orientation, so to be accurately aligned with the input subset. Otherwise, that input subvolume should be discarded or its location and size should be updated until a valid object is found. In the meantime, once a valid object is found, the network should be able to detect all other similar objects accordingly, instead of needing excessive search. To achieve this goal, we introduce two parallel branches for the network, 1) an *object discovery branch* which is innovatively formulated as an embodied agent to explore and interact with 3D scene point clouds by reinforcement learning (RL), while receiving rewards from our pretrained generative object-centric network, and 2) an *object segmentation branch* supervised by pseudo object labels discovered. Notably, the embodied agent based discovery branch is discarded once the segmentation branch is well trained, thus the whole multi-object estimation network is efficient during inference.

Our framework, named **GrabS**, learns **g**ene**r**ative object priors via the object-centric network and trains an embodied **a**gent to discover o**b**jects, ultimately allowing us to effectively **s**egment multiple objects on scene point clouds. The closest work to us is EFEM (Lei et al., 2023), but its learned object priors are not generative and the object discovery strategy heavily relies on heuristics to search limited objects. Our contributions are:

- We introduce a two-stage learning pipeline for 3D object segmentation. An object-centric generative model is designed to learn both discriminative and robust object priors.
- We present a multi-object estimation network to effectively discover individual objects by training a newly designed embodied agent which interacts with 3D scenes and receives rewards from the pretrained generative object-centric priors without needing human labels in training.
- We demonstrate superior object segmentation results and clearly surpass baselines on multiple datasets. Our code and data are available at `https://github.com/vLAR-group/GrabS`

## 2 RELATED WORKS

**Fully-/Weakly-supervised 3D Object Segmentation**: Significant progress has been made in fully-supervised object segmentation of 3D point clouds, including bottom-up point clustering methods (Wang et al., 2018; Han et al., 2020; Chen et al., 2021; Vu et al., 2022), top-down detection based approaches (Hou et al., 2019; Yi et al., 2019; Yang et al., 2019; He et al., 2021; Shin et al., 2024), and Transformer based methods (Jiahao Lu et al., 2023; Lai et al., 2023; Schult et al., 2023; Sun et al., 2023; Kolodiazhnyi et al., 2024). A number of succeeding methods leverage relatively

weak labels such as 3D bounding boxes (Chibane et al., 2022; Tang et al., 2022) or object centers (Griffiths et al., 2020) to identify 3D objects. Although achieving excellent accuracy on public 3D datasets, they primarily rely on laborious human annotations to train neural networks.

**3D Object Segmentation with Self-supervised/Supervised 2D/3D Features**: Recently, with the advancement of self-supervised pretraining techniques and fully-supervised foundation models, a line of methods (Ha & Song, 2022; Lu et al., 2023; Takmaz et al., 2023; Liu et al., 2023; Nguyen et al., 2024; Yan et al., 2024; Roh et al., 2024; Boudjoghra et al., 2024; Tai et al., 2024; Yin et al., 2024) have been developed to leverage pretrained 2D/3D or language features (Xie et al., 2020; Caron et al., 2021; Radford et al., 2021; Kirillov et al., 2023) as supervision signals to discover 3D objects on closed or open world datasets. Despite showing promising results, these methods still rely on large-scale annotated data in 2D/3D domain or aligned vision-language data pairs, making them costly and unappealing in real applications.

**Unsupervised 3D Object Segmentation**: To avoid data annotation, a couple of recent methods are proposed to discover 3D objects by relying on heuristics such as distributions of point normals/colors/motions (Zhang et al., 2023; 2024; Song & Yang, 2022; 2024), or the similarity of pretrained features from 2D domain (Rozenberszki et al., 2024; Shi et al., 2024). However, their capability of identifying complex 3D objects is often inferior.

**3D Object-centric Prior Learning**: To learn object-centric priors, most methods usually train a deterministic reconstruction network to predict 3D objects in different representations such as mesh (Yang et al., 2018), point clouds (Fan et al., 2017), signed distance fields (SDF) (Park et al., 2019), and unsigned distance fields UDF (Chibane et al., 2020), whereas another line of works (Achlioptas et al., 2018; Kim et al., 2021; Klokov et al., 2020; Luo & Hu, 2021; Chou et al., 2023; Li et al., 2023a; Zeng et al., 2022) train a generative network to learn object shape distributions using Generative Adversarial Networks (GAN) (Goodfellow et al., 2014), VAE (Kingma & Welling, 2014), normalizing flows (Kim et al., 2020), or diffusion models (Ho et al., 2020). These methods often aims to generate a diverse range of single 3D objects. By contrast, our framework is not primarily targeting at 3D generation, but demonstrating the ability to discover multiple 3D objects.

## 3 GRABS

### 3.1 OVERVIEW

Our framework consists of two stages/networks. The object-centric network is designed to learn object-level generative priors from a set of individual 3D objects (*e.g.*, thousands of chairs in ShapeNet (Chang et al., 2015)). After this object-centric network is well trained and frozen, our ultimate goal is to use it to optimize another multi-object estimation network to discover as many similar objects as possible on complex 3D scene point clouds such as thousands of 3D rooms in ScanNet (Dai et al., 2017). Details of the two networks are discussed in Sections 3.2&3.3.

### 3.2 OBJECT-CENTRIC NETWORK

Given a set of individual 3D objects usually with a canonical pose collected in existing datasets, a specific object is denoted as $\boldsymbol{O} \in \mathcal{R}^{M \times 3}$ where $M$ represents the number of 3D points with three channels of location $xyz$. Other possible features such as RGB or normals are ignored in this paper for simplicity. Our object-centric network comprises two modules elaborated below and they will be trained on these 3D objects.

**Object Orientation Estimation Module**: Our final goal is to segment potential objects in 3D scene point clouds, but those objects are usually located with unknown poses. This means that our object-centric network should be able to first infer various orientations of objects with respect to a canonical pose. To this end, we introduce a neural module $f_R$ to directly regress the orientation of an input object point cloud or the inverse of the viewing angle with regard to a canonical pose.

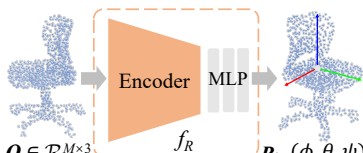

Figure 2: Object orientation estimation module.

As illustrated in Figure 2, given an object point cloud $\boldsymbol{O}$, we feed it into an encoder to obtain a 128-dimensional global vector followed by multilayer perceptrons (MLPs) to directly regress three

Figure 3: The object generative prior learning module with VAE-based and diffusion-based variants.

orientation parameters $\boldsymbol{R} \leftarrow (\phi, \theta, \psi)$. For simplicity, we adopt PointNet++ (Qi et al., 2017) with a self-attention layer in each block as our encoder, though other sophisticated backbones can also be used, and the L1 loss is applied following (Ke et al., 2020). To train this module, we create a sufficient number of training pairs by randomly rotating canonically-posed synthetic objects in ShapeNet. Details of the neural architecture and dataset preparation are in Appendix A.

**Object Generative Prior Learning Module**: Again, our final objective is to identify multiple objects in 3D scenes, but those objects often come with noises, self or mutual occlusions, and/or domain shifts. A naïve solution is to augment existing object-level datasets by creating countless samples for training, but this is data inefficient. We argue that it is more preferable to learn a generative module $f_G$, as it is inherently capable of capturing more robust and continuous latent distributions from a moderate amount of 3D objects, as also validated by our ablation in Section 4.4.

As shown in the left block of Figure 3, we adopt a VAE framework (Kingma & Welling, 2014) to learn conditional latent distributions. In particular, this module takes a canonically-posed object point cloud $\bar{\boldsymbol{O}}$ as input to an encoder, learning a 128-dimensional latent distribution $\mathcal{N}(\boldsymbol{\mu}, \boldsymbol{\sigma^2})$. The encoder architecture is the same as our object orientation estimation module. The sampled latent code is then fed into MLPs to learn an SDF (Park et al., 2019). This SDF decoder exactly follows EFEM (Lei et al., 2023). As shown in the right block of Figure 3, our module is also flexible to adopt the popular diffusion model as an alternative. Following Diffusion-SDF (Chou et al., 2023), our diffusion-based variant learns to denoise latent codes and is trained jointly with our VAE variant. All details of the encoder/decoder and VAE/diffusion layers are provided in Appendix A.

**Training & Test**: Both our object orientation estimation module $f_R$ and object generative prior learning module $f_G$ are simply trained in a fully-supervised manner on an object dataset. Since our final goal is to leverage the pretrained $f_R$ and $f_G$ for multi-object segmentation, it is less important to test its ability of generating high-quality single objects on benchmarks.

### 3.3 MULTI-OBJECT ESTIMATION NETWORK

With the object-centric network well-trained on an object dataset, our core objective is to segment many similar objects on complex scene point clouds without human labels for training. Given a single scene point cloud, a naïve solution is to randomly crop many subvolumes of points at different locations with different volume sizes, and then feed these subvolumes into our pretrained object-centric network, obtaining their orientations followed by shape reconstruction. By verifying whether each subvolume has a set of points that can be reconstructed, we can regard the perfectly reconstructed point sets as discovered objects. However, such a random cropping is extremely inefficient due to the lack of a suitable learning strategy. In fact, it is also infeasible to directly learn subvolume parameters like regressing object bounding boxes, essentially because the cropping operation is non-differentiable. To this end, we introduce a novel multi-object estimation network to discover objects by an embodied agent. The network has two parallel branches sharing a backbone.

Given an input scene point cloud $\boldsymbol{P} \in \mathcal{R}^{N \times 3}$, we first feed it into a backbone network (not pretrained) $f_{bone}$, obtaining per-point features $\boldsymbol{F} \in \mathcal{R}^{N \times 128}$ which will be used in our two branches as discussed below. SparseConv (Graham et al., 2018) is adopted as the backbone for simplicity.

**Object Discovery Branch as an Embodied Agent**: Due to the lack of human labels in training, discovering objects is actually a trial-and-error process. In this regard, we formulate it as an embodied agent to actively search objects via RL, which also does not require dense or continuous labels. Because our pretrained object-centric network inherently has rich object priors and can serve as an indicator of objectness, it is naturally suitable to act as a reward generator. We set up the embodied agent learning pipeline as follows, including definitions of the agent and its action space, a policy network, reward design, and the training loss.

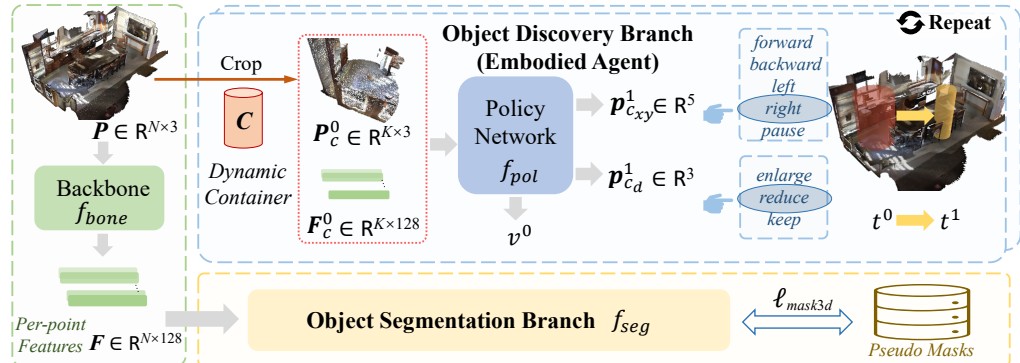

Figure 4: The framework of multi-object estimation network.

**1) Embodied Agent**: An ambitious agent is to directly discover object masks in action space, but the exploration cost would exponentially grow against the number of 3D scene points. In this regard, we opt to learn fewer parameters instead. Particularly, we create an agent called *dynamic container* $C$ in 3D scene space. For simplicity, we choose it as a cylinder with an unconstrained height and parameterize it by its center projected on $xy$ plane and its diameter, *i.e.*, $C \leftarrow [C_x, C_y, C_d]$. This agent is expected to start from an initial size and random location, then dynamically change its parameters in its action space according to a policy network, and receive rewards by querying against our pretrained object-centric network, eventually moving to a valid object.

**2) Action Space**: Regarding the three parameters $[C_x, C_y, C_d]$, we define two groups of actions as follows. To speed up exploration, both groups are executed simultaneously at every timestamp.

- Group #1: The dynamic container will move {*forward, backward*} to update $C_x$, move {*left, right*} to update $C_y$ all by a fixed and predefined step size $\Delta s$, or keep still with an action *pause*. During exploration, the dynamic container will only choose one of the five actions at every timestamp to update its two location parameters $[C_x, C_y]$.

- Group #2: The dynamic container will {*enlarge, reduce*} its diameter $C_d$ by a fixed and predefined ratio $\alpha$ to update its current volume size, or keep unchanged with an action *keep*. During exploration, the container will only choose one of the three actions at every timestamp to adjust its size parameter $C_d$.

**3) Policy Network**: As shown in the leftmost block of Figure 4, we have per-point features $F$ of the input scene $P$. Assuming the container agent has randomly initialized parameters $[C_x^0, C_y^0, C_d^0]$ at time $t^0$, we crop the corresponding 3D points and features within container, denoted as $P_c^0 \in \mathcal{R}^{K \times 3}$ and $F_c^0 \in \mathcal{R}^{K \times 128}$ respectively, where $K$ represents the number of points and may vary at future timestamps. Both $P_c^0$ and $F_c^0$ are regarded as container state features at time $t_0$, and they are fed into our attention-based policy network $f_{pol}$, directly predicting a policy for next actions of both groups at time $t^1$, denoted as $p_{c_{xy}}^1 \in \mathcal{R}^5$ and $p_{c_d}^1 \in \mathcal{R}^3$ respectively. We also estimate a state value $v^0$ in parallel, as illustrated by the upper block of Figure 4. With the predicted policy, the agent will execute corresponding actions at next timestamps, and the future exploration will repeat this process until the agent being stopped. Note that, over training, the container's future steps will be more likely to approach a valid object, though its first step is always randomly initialized.

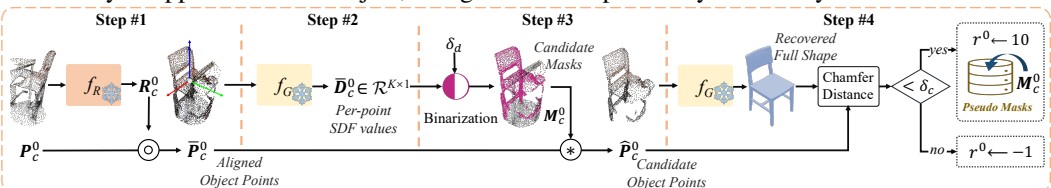

Figure 5: The steps to generate rewards for the container from our pretrained object-centric network.

**4) Reward Design via Pretrained Object-centric Network**: At time $t^0$, given the container state features $P_c^0$, *i.e.*, a subset of points cropped, we will query it against our object-centric network pretrained in Section 3.2, obtaining a reward $r^0$ following the steps below and shown in Figure 5.

- Step #1: The subset of points $P_c^0$ is first fed into the pretrained object orientation module $f_R$, getting its pose $R_c^0$. This subset is then aligned to a canonical pose following $\bar{P}_c^0 \leftarrow P_c^0 \circ R_c^0$.

- Step #2: The aligned point subset $\bar{\boldsymbol{P}}_c^0$ is fed into the pretrained object generative prior module $f_G$. By querying these $K$ points through the SDF decoder, we will obtain the corresponding per-point distance values $\bar{\boldsymbol{D}}_c^0 \in \mathcal{R}^{K \times 1}$.

- Step #3: We will compute a candidate object mask $\boldsymbol{M}_c^0 \in \mathcal{R}^{K \times 1}$ by binarizing the absolute distance values of $\bar{\boldsymbol{D}}_c^0$ against a predefined threshold $\delta_d$. For those points whose absolute distance values are smaller than $\delta_d$, they are regarded as a candidate object surface points. The candidate object points are physically carved out (denoted by an operation *): $\hat{\boldsymbol{P}}_c^0 \leftarrow \bar{\boldsymbol{P}}_c^0 * \boldsymbol{M}_c^0$.

- Step #4: Lastly, we will feed this object points $\hat{\boldsymbol{P}}_c^0$ into our pretrained object generative prior module $f_G$ and reconstruct a full object shape via Marching Cubes on the SDF decoder. If the Chamfer distance between the input candidate object $\hat{\boldsymbol{P}}_c^0$ and the recovered full shape (sampled dense points) is smaller than a threshold $\delta_c$, the reward $r^0$ is assigned with a positive score, *e.g.*, $r^0 \leftarrow 10$, a negative score otherwise, *e.g.*, $r^0 \leftarrow -1$. Notably, for the candidate object with a positive score, its mask will always be stored in an external list and regarded as a pseudo object mask to train our object segmentation branch.

**5) Training Loss**: For the dynamic container, given its initial states: $\boldsymbol{P}_c^0$ and $\boldsymbol{F}_c^0$ obtained from backbone $f_{bone}$ at time $t^0$, we have its predicted policy and state value: $\{\boldsymbol{p}_{c_{xy}}^1, \boldsymbol{p}_{c_d}^1, v^0\}$ through policy network $f_{pol}$, and a reward $r^0$ via our pretrained object-centric network. By executing actions according to the predicted policy, we collect a sufficient number of trajectories to optimize both the backbone and the policy network using an existing PPO loss (Schulman et al., 2017).

$$(f_{bone}, f_{pol}) \longleftarrow \ell_{ppo} \tag{1}$$

Thanks to the learned object-centric priors and our creative embodied agent based formulation of object detection, this object discovery branch can successfully identify multiple objects from complex scene point clouds. In implementation, we divide 3D scenes into smaller blocks for the container to search within them in parallel, thus speeding up the exploration. Details of agent initialization, policy network, rewards, loss functions, parallelization, and hyperparameters are in Appendix K.

**Object Segmentation Branch**: Given the input scene point cloud $\boldsymbol{P}$, during the exploration of dynamic container, we will have a list of object masks accumulated as pseudo labels. Clearly, these objects are valuable for us to directly train a segmentation branch, thus similar objects are more likely to be detected even if they may be missed by our agent of dynamic container. To this end, as illustrated in the lower block of Figure 4, our object segmentation branch $f_{seg}$ takes per-point features $\boldsymbol{F}$ as input and then exactly follows Mask3D (Schult et al., 2023) to directly predict a set of class-agnostic object masks for the entire input point cloud $\boldsymbol{P}$. The existing supervision loss from Mask3D (Schult et al., 2023) is applied to optimize both backbone and the segmentation branch. More details of the neural architecture, loss functions, and training settings are in Appendix L.

$$(f_{bone}, f_{seg}) \longleftarrow \ell_{mask3d} \tag{2}$$

Overall, our framework GrabS first learns an object-centric network for object pose alignment followed by generative shape prior learning on existing object-level datasets. With the learned priors as a foundation, we introduce a novel multi-object estimation network to segment multiple individual objects from complex 3D scene point clouds without needing human labels to train.

## 4 EXPERIMENTS

**Datasets**: We evaluate on three datasets: 1) The challenging real-world **ScanNet** dataset (Dai et al., 2017), comprising 1201/312/100 indoor scenes for training/validation/test respectively; 2) The real-world **S3DIS** dataset (Armeni, 2017), including 6 areas of indoor scenes; 3) Our own **synthetic dataset** with 4000/1000 training/test scenes. Following EFEM (Lai et al., 2023), we first train the object-centric network on ShapeNet and then conduct object segmentation on scene datasets.

**Baselines**: We compare with the following relevant methods. 1) **EFEM** (Lei et al., 2023): it is the closest work to us, which also learns object priors from ShapeNet and then segments objects without needing scene annotations in training. 2) **EFEM**$_{mask3d}$: we further build this baseline by training a Mask3D model using the discovered pseudo labels from EFEM. This model maintains the same architecture and train settings as our object segmentation branch. 3) **Unscene3D** (Rozenberszki et al., 2024): it is an unsupervised 3D object segmentation method which leverages 2D pre-trained

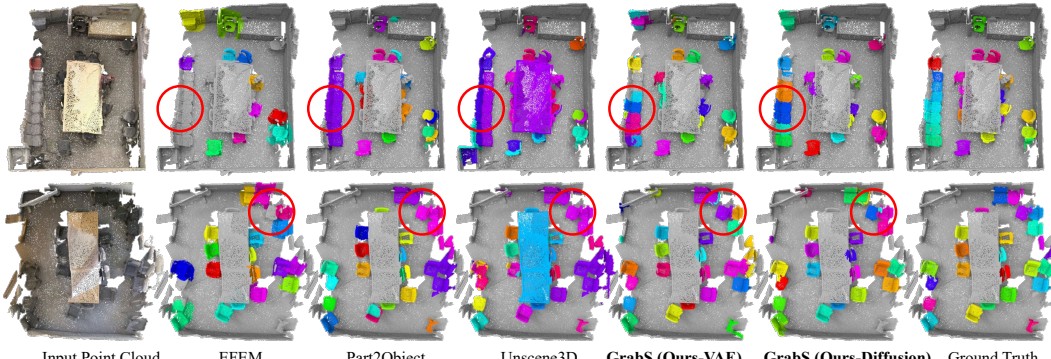

Figure 6: Qualitative results on the ScanNet validation set. Red circles highlight the differences.

DINO features to provide pseudo 3D annotations to train a detector. 4) **Part2Object** (Shi et al., 2024): it is an unsupervised method also incorporating DINO features. For reference, we also include the 3D fully supervised method **Mask3D** (Schult et al., 2023), and the recent 2D/language fully supervised models **OpenIns3D** (Huang et al., 2024) and **SAI3D** (Yin et al., 2024).

Regarding our pipeline, once the object discovery branch is well-trained by RL, the policy network alone can discover multiple objects on point clouds by querying the frozen object-centric network (*e.g.*, our VAE version) in testing. Intuitively, given more trajectories, the object discovery branch is likely to identify more objects. For a comparison, we directly test our well-trained object discovery branch, given 50/100/300/600 trajectories respectively, denoted as GrabS(Ours-VAE)$_{dis50}$. Note that, the original EFEM uses 600 trajectories on each scene when discovering objects.

**Metrics**: Evaluation metrics include the standard average precision (AP), recall (RC), and precision (PR) scores at different IoU thresholds.

## 4.1 EVALUATION ON SCANNET

Following the settings of EFEM (Lei et al., 2023) for a fair comparison, we train an object-centric network on the chair category of ShapeNet and subsequently train a multi-object estimation network in an unsupervised manner on the training set of ScanNet. We evaluate the performance exclusively on chairs in both validation and the online hidden test sets, treating all our predicted masks as chairs.

**Results & Analysis:** Table 1 and Figure 6 present the quantitative and qualitative results. We can see that: 1) Our method significantly outperforms the closest work EFEM (Lai et al., 2023). 2) For the other two unsupervised methods Unscene3D (Rozenberszki et al., 2024) and Part2Object (Shi et al., 2024), we assign ground truth class labels to their predicted masks and exclude all non-chair predictions. We clearly surpass them on all metrics, demonstrating the superiority of our method.

Table 2 compares the results on the hidden test set of ScanNet. It can be see that our method is significantly better than EFEM and achieves closing scores to an early 3D fully-supervised method 3D-BoNet (Yang et al., 2019), showing great potential of our unsupervised learning scheme. Nevertheless, we also notice that our method has a performance gap between validation and hidden test sets. We hypothesize that this is likely caused by a distribution gap between two sets, because the fully supervised method Mask3D also shows a clear performance gap on two sets. With more 3D object and scene datasets collected in the future, we believe the domain gap can be narrowed down. More qualitative results are provided in Appendix N.

## 4.2 EVALUATION ON S3DIS

Similar to the ScanNet dataset, we only evaluate on the chair category. For a fair comparison, we exactly follow the existing two unsupervised methods Unscene3D (Rozenberszki et al., 2024) and Part2Object (Shi et al., 2024) to conduct cross dataset validation on S3DIS. In particular, we directly use our multi-object estimation network trained on the ScanNet dataset in Section 4.1 to evaluate on the test set of S3DIS. Note that, the baseline EFEM does not have a training stage and it is directly applied on the test set of S3DIS according to its own design.

Table 1: Quantitative results of our method and baselines on the validation set of ScanNet.

| | | AP(%) | AP50(%) | AP25(%) | RC(%) | RC50(%) | RC25(%) | PR(%) | PR50(%) | PR25(%) |
|---|---|---|---|---|---|---|---|---|---|---|
| 3D Supervised | Mask3D (Schult et al., 2023) | 82.9 | 94.4 | 97.0 | - | - | - | - | - | - |
| 2D Foundation | OpenIns3D (Huang et al., 2024) | 66.7 | 82.4 | 85.7 | - | - | - | - | - | - |
| Model Supervised | SAI3D (Yin et al., 2024) | 38.5 | 62.5 | 81.2 | 54.3 | 79.9 | 95.4 | 38.1 | 56.4 | 70.1 |
| | Unscene3D (Rozenberszki et al., 2024) | 37.2 | 62.4 | 79.2 | 51.7 | 70.4 | 84.1 | 18.7 | 26.3 | 29.7 |
| | Part2Object (Shi et al., 2024) | 34.4 | 56.8 | 73.9 | 46.4 | 65.5 | 78.5 | 45.5 | 65.4 | 76.7 |
| | EFEM (Lei et al., 2023) | 24.6 | 50.8 | 61.3 | - | - | - | - | - | - |
| Unsupervised | $EFEM_{mask3d}$ | 38.8 | 55.1 | 63.8 | 52.4 | 68.7 | 80.8 | 18.8 | 27.1 | 29.1 |
| | GrabS (Ours-VAE)$_{dis50}$ | 26.3 | 50.7 | 56.9 | 36.0 | 54.3 | 60.6 | **55.3** | **85.3** | **93.3** |
| | GrabS (Ours-VAE)$_{dis100}$ | 26.9 | 51.2 | 59.1 | 35.6 | 53.9 | 60.3 | 53.6 | 82.8 | 91.3 |
| | GrabS (Ours-VAE)$_{dis300}$ | 28.5 | 55.2 | 66.8 | 39.3 | 60.8 | 69.5 | 46.5 | 73.9 | 82.5 |
| | GrabS (Ours-VAE)$_{dis600}$ | 28.7 | 56.2 | 66.9 | 39.5 | 61.6 | 69.5 | 45.9 | 73.7 | 81.5 |
| | GrabS (Ours-VAE) | 46.7 | **71.5** | **82.9** | **53.2** | **74.5** | **85.2** | 52.1 | 76.4 | 83.0 |
| | **GrabS (Ours-Diffusion)** | **47.1** | 70.6 | 81.1 | 52.9 | 73.3 | 82.9 | 54.9 | 79.2 | 85.7 |

Table 2: Quantitative results of our method and baselines on the hidden test set of ScanNet.

| | | AP(%) | AP50(%) | AP25(%) |
|---|---|---|---|---|
| 3D Supervised | 3D-BoNet (Yang et al., 2019) | 34.5 | 48.4 | 64.3 |
| | SoftGroup (Vu et al., 2022) | 69.4 | 86.2 | 91.3 |
| | Mask3D (Schult et al., 2023) | 73.7 | 88.5 | 93.8 |
| Unsupervised | EFEM (Lei et al., 2023) | 20.2 | 39.0 | 48.3 |
| | **GrabS (Ours-VAE)** | **29.0** | **45.1** | 57.7 |
| | GrabS (Ours-Diffusion) | 28.5 | 43.1 | **58.1** |

**Results & Analysis:** As shown in Tables 3&4, our method significantly outperforms all unsupervised baselines overall, though both Unscene3D and Part2Object distill particularly strong visual features from the well-trained and powerful DINO model. Upon a closer look at the qualitative results shown in Figure 7, we can find that: 1) EFEM is likely to miss detecting objects, primarily because its object discovery stage relies on heuristics instead of learning a general detector like ours. 2) Both Unscene3D and Part2Object are struggling to separate similar objects near each other, or likely to oversegment objects into parts. This is mainly because the pretrained 2D DINO features do not capture nuanced object-centric representations, though those features have hints of object locations and rough shapes. By contrast, our method learns discriminative and robust 3D object-centric priors which give the multi-object estimation network precise signals to identify objects. More quantitative and qualitative results are in Appendix O.

Table 3: Quantitative results of cross dataset validation on the Area-5 of S3DIS.

| | | AP(%) | AP50(%) | AP25(%) | RC(%) | RC50(%) | RC25(%) | PR(%) | PR50(%) | PR25(%) |
|---|---|---|---|---|---|---|---|---|---|---|
| | Unscene3D (Rozenberszki et al., 2024) | 42.6 | 63.4 | **80.3** | **51.9** | **68.6** | **83.3** | 17.4 | 23.5 | 27.7 |
| | Part2Object (Shi et al., 2024) | 30.0 | 50.5 | 76.4 | 45.2 | 64.7 | 82.6 | 43.5 | 62.5 | 80.1 |
| Unsupervised | EFEM (Lei et al., 2023) | 14.9 | 35.7 | 45.3 | 18.6 | 36.0 | 45.3 | 43.6 | 92.1 | **100.0** |
| | **GrabS (Ours-VAE)** | **46.4** | **66.2** | 73.8 | 51.0 | 67.1 | 74.0 | 68.5 | 91.5 | 97.0 |
| | GrabS (Ours-Diffusion) | 44.2 | 58.0 | 62.6 | 45.7 | 58.9 | 63.2 | **70.8** | **91.6** | 96.4 |

## 4.3 EVALUATION ON A SYNTHETIC DATASET

Although we conduct experiments only on the chair category in Sections 4.1&4.2 for fair comparisons with baselines, the design of our method is agnostic to any object categories. To further evaluate the effectiveness of our method on discovering multiple classes of objects by a single network, we choose to create a synthetic room dataset using objects from ShapeNet.

In particular, we create 4000/1000 3D indoor rooms (scenes) for training/test respectively. To avoid data leakage, for each training scene, we select 3D objects only from the *validation set* of ShapeNet, whereas for each test scene, we select 3D objects only from the *test set* of ShapeNet. In each training/test room (scene), we randomly place 4~8 objects belonging to 6 classes of ShapeNet {chair, sofa, telephone, airplane, rifle, cabinet}. Note that, we only use 3D objects of the above 6 classes in the *training set* of ShapeNet to train a single object-centric network. More details about our synthetic dataset are provided in Appendix M, and we have released it for future studies.

To conduct class-agnostic object segmentation on our synthetic dataset, we include the classic algorithm HDBSCAN (McInnes & Healy, 2017) as an unsupervised baseline in addition to EFEM. For Unscene3D and Part2Object, both require paired RGB images to extract 2D features via pretrained DINO/v2 for training their own detection networks, so it is unable to directly train them on our synthetic dataset due to the lack of paired RGB images. For reference, we directly reuse their models well-trained on the training set of ScanNet in Section 4.1, and then test on our synthetic dataset. Since such a setting is not strictly fair to them, we group them as the category "Unsupervised&Real2Syn". For reference, we also train a fully supervised Mask3D (Schult et al., 2023).

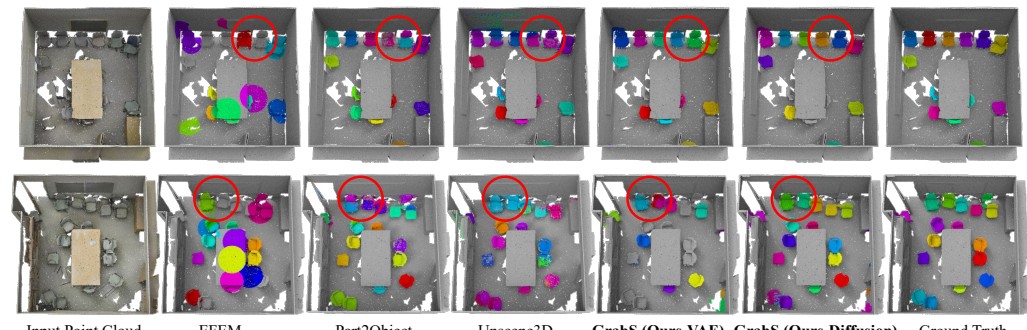

| Input Point Cloud | EFEM | Part2Object | Unscene3D | **GrabS (Ours-VAE)** | **GrabS (Ours-Diffusion)** | Ground Truth |

Figure 7: Qualitative results on the S3DIS dataset. Red circles highlight the differences.

Table 4: Quantitative results of cross dataset validation on all 6 Areas of S3DIS.

| | | AP(%) | AP50(%) | AP25(%) | RC(%) | RC50(%) | RC25(%) | PR(%) | PR50(%) | PR25(%) |
|---|---|---|---|---|---|---|---|---|---|---|
| | Unscene3D (Rozenberszki et al., 2024) | 30.3 | 51.9 | **70.4** | 40.0 | 58.6 | **73.9** | 13.8 | 19.7 | 24.5 |
| | Part2Object (Shi et al., 2024) | 25.3 | 48.4 | 67.0 | 36.5 | 57.3 | 72.3 | 37.8 | 60.5 | 76.4 |
| Unsupervised | EFEM (Lei et al., 2023) | 16.2 | 37.8 | 45.9 | 20.5 | 40.4 | 45.5 | 41.5 | 76.1 | **99.6** |
| | **GrabS (Ours-VAE)** | **41.8** | **61.7** | 67.0 | **45.9** | **63.0** | 67.9 | 60.0 | **84.0** | 90.7 |
| | GrabS (Ours-Diffusion) | 39.2 | 57.2 | 62.6 | 42.2 | 58.2 | 63.0 | **60.0** | 73.7 | 79.7 |

Table 5: Quantitative results on the test set of our synthetic dataset.

| | | AP(%) | AP50(%) | AP25(%) | RC(%) | RC50(%) | RC25(%) | PR(%) | PR50(%) | PR25(%) |
|---|---|---|---|---|---|---|---|---|---|---|
| 3D Supervised | Mask3D (Schult et al., 2023) | 84.1 | 96.0 | 96.7 | 87.1 | 96.2 | 96.9 | 89.5 | 98.9 | 99.5 |
| Unsupervised & Real2Syn | Unscene3D (Rozenberszki et al., 2024) | 37.7 | 59.7 | 76.2 | 50.5 | 70.4 | 83.9 | 8.2 | 14.8 | 15.4 |
| | Part2Object (Shi et al., 2024) | 46.1 | 69.3 | 81.5 | 53.1 | 70.9 | 83.4 | 10.2 | 15.0 | 19.1 |
| | HDBSCAN (McInnes & Healy, 2017) | 7.6 | 12.5 | 23.4 | 10.8 | 15.8 | 24.7 | 36.6 | 58.5 | 90.0 |
| Unsupervised | EFEM (Lei et al., 2023) | 20.7 | 34.1 | 46.6 | 23.3 | 34.7 | 46.7 | 53.3 | 90.6 | **98.7** |
| | **GrabS (Ours-VAE)** | **58.7** | 85.0 | 90.6 | 71.6 | 87.9 | 91.1 | 76.0 | 93.7 | 96.3 |
| | GrabS (Ours-Diffusion) | 58.5 | **85.9** | **91.5** | **72.4** | **88.7** | **91.7** | **77.9** | **95.7** | 98.5 |

**Results & Analysis:** As shown in Table 5 and Figure 8, our method clearly outperforms HDBSCAN and EFEM by a large margin, because HDBSCAN can hardly group points into complex 3D shapes and EFEM can only detect limited objects based on its heuristics. More results are in Appendix P.

## 4.4 ABLATION STUDY

To evaluate the effectiveness of each component of our pipeline and the choices of hyperparameters, we conducted the following extensive ablation experiments on the validation set of ScanNet. We choose the VAE version of object-centric network as our full framework for reference.

**(1) Using a deterministic object-centric network**. This is to assess the advantages of learning generative object-centric priors. In particular, we simply replace the probabilistic latent distributions of VAE by a deterministic latent vector (AE), keeping other layers of our object-centric network unchanged. After training such a deterministic network, we then use it to optimize our multi-object estimation network on 3D scenes as the same as our full framework.

**(2) Removing the Object Orientation Estimation Module**. This is to evaluate the importance of aligning an input object point cloud with respect to a canonical pose. Without it, the chaotic object orientations in complex 3D scenes may cause the performance drop of object segmentation.

**(3) Removing the Object Discovery Branch**. This is to evaluate the effectiveness of embodied agent for object discovery. Without it, we randomly select 50 positions in each scene and randomly set a radius ranging from $0 \sim 2$ meters as the container for discovering objects as pseudo labels.

**(4) Removing the Object Segmentation Branch**. This is to evaluate the effectiveness of the Object Segmentation Branch. Without it, we only use the Object Discovery Branch to collect object masks by querying against the frozen object-centric network.

**(5) $\sim$ (8) Sensitivity to the container position moving step $\Delta s$.** This aims to evaluate the influence of different choices of moving step $\Delta s$ when the dynamic container is exploring the 3D space.

**(9) $\sim$ (11) Sensitivity to the container size changing ratio $\alpha$.** This aims to evaluate the influence of different choices of container size varying speed $\alpha$ when it is exploring the 3D space.

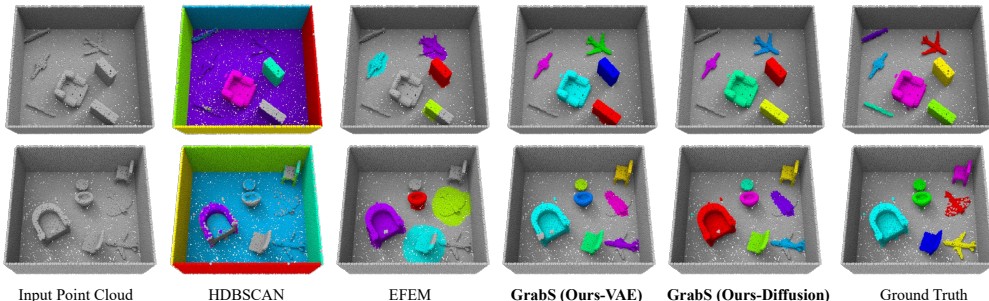

| Input Point Cloud | HDBSCAN | EFEM | **GrabS (Ours-VAE)** | **GrabS (Ours-Diffusion)** | Ground Truth |

Figure 8: Qualitative results on the test set of our synthetic dataset.

**(12) ∼ (15) Sensitivity to the binary threshold** $\delta_d$. The threshold $\delta_d$ helps to convert SDF values of surface points into a binary object mask, which mainly influences the quality of a pseudo mask.

**(16) ∼ (18) Sensitivity to the reward threshold** $\delta_c$. Since rewards are critical for the agent, we therefore conduct four ablations on the threshold $\delta_c$, which verifies whether the input point cloud is matched with its reconstructed 3D shape and then assigns positive or negative rewards.

Table 6: Results of all ablated models of our GrabS on the ScanNet validation set. The bold settings are chosen in our full framework.

| | AP(%) | AP50(%) | AP25(%) | RC(%) | RC50(%) | RC25(%) | PR(%) | PR50(%) | PR25(%) |
|---|---|---|---|---|---|---|---|---|---|
| (1) Replace VAE by AE | 32.0 | 57.1 | 76.7 | 46.2 | 73.7 | 90.6 | 26.2 | 45.9 | 52.6 |
| (2) Remove Orientation Estimator | 35.3 | 56.3 | 69.7 | 42.9 | 61.5 | 72.0 | 52.4 | 77.6 | 87.5 |
| (3) Remove Object Discovery Branch | 34.2 | 56.7 | 69.4 | 41.6 | 61.4 | 73.1 | 50.7 | 78.1 | 89.0 |
| (4) Remove Object Segmentation Branch | 25.7 | 47.9 | 55.3 | 33.5 | 50.2 | 56.0 | 57.1 | 87.1 | 95.5 |
| (5) $\Delta s = 0.2$ | 43.6 | 61.3 | 72.0 | 48.0 | 63.1 | 71.6 | 62.3 | 84.1 | 91.3 |
| **(6) $\Delta s = 0.3$** | **46.7** | **71.5** | **82.9** | **53.2** | **74.5** | **85.2** | 52.1 | 76.4 | 83.0 |
| (7) $\Delta s = 0.4$ | 40.1 | 58.1 | 67.4 | 44.9 | 60.8 | 69.6 | 57.1 | 79.5 | 87.3 |
| (8) $\Delta s = 0.5$ | 38.1 | 56.5 | 66.9 | 43.4 | 59.8 | 69.9 | 51.8 | 73.7 | 82.6 |
| (9) $\alpha = 1/3$ | 40.8 | 61.4 | 72.9 | 48.0 | 66.4 | 77.3 | 48.8 | 71.6 | 77.7 |
| **(10) $\alpha = 1/4$** | **46.7** | **71.5** | **82.9** | **53.2** | **74.5** | **85.2** | 52.1 | 76.4 | 83.0 |
| (11) $\alpha = 1/5$ | 43.3 | 61.9 | 69.5 | 48.4 | 64.5 | 71.7 | 59.9 | 81.8 | 87.2 |
| (12) $\delta_d = 0.01$ | 40.1 | 59.3 | 67.8 | 45.5 | 61.7 | 69.1 | 61.2 | 85.1 | 91.9 |
| **(13) $\delta_d = 0.02$** | **46.7** | **71.5** | **82.9** | **53.2** | **74.5** | **85.2** | 52.1 | 76.4 | 83.0 |
| (14) $\delta_d = 0.05$ | 43.4 | 62.8 | 69.3 | 47.8 | 64.0 | 69.8 | 64.9 | 88.4 | 94.0 |
| (15) $\delta_d = 0.10$ | 42.9 | 61.4 | 69.5 | 47.2 | 62.4 | 70.1 | 63.2 | 86.0 | 93.7 |
| (16) $\delta_c = 0.12$ | 42.6 | 60.8 | 67.6 | 47.6 | 63.7 | 69.9 | 63.1 | 85.7 | 91.2 |
| **(17) $\delta_c = 0.14$** | **46.7** | **71.5** | **82.9** | **53.2** | **74.5** | **85.2** | 52.1 | 76.4 | 83.0 |
| (18) $\delta_c = 0.16$ | 43.8 | 68.8 | 81.0 | 52.1 | 75.8 | 86.9 | 41.3 | 63.7 | 68.6 |

**Analysis:** From Table 6, we can see that: 1) The choice of learning generative object-centric priors has the greatest impact on our framework. Without it, the AP score drops significantly. This is because the deterministic shape priors are not robust and continuous in latent space, thus being unable to generalize to real-world 3D scenes where objects are vastly different from ShapeNet objects. 2) The removal of the Object Orientation Estimation module shows the next greatest impact on performance, demonstrating that this module is necessary to align orientations of real-world objects. 3) For the four hyperparameters in our multi-object estimation network, the overall performance is not easily affected too much by different choices, showing the robustness of our framework. More ablations about $\delta_c$ on S3DIS are in Appendix E. During discovering objects by the embodied agent, we create multiple trajectories in parallel by dividing a 3D scene into smaller blocks. We further conduct ablation experiments on the number of parallel trajectories in Appendix C.

## 5 CONCLUSION

In this paper, we have shown that multiple 3D objects can be effectively discovered from complex real-world point clouds without needing human labels of 3D scenes in training. This is achieved by our new two-stage learning pipeline, where the first stage learns generative object priors by an object-centric network on large-scale object datasets. By querying against the learned priors and receiving rewards of objectness, the second stage learns to discover similar 3D objects via a newly formulated embodied agent in our multi-object estimation network. Extensive experiments on multiple real-world datasets and our created synthetic dataset have demonstrated the excellent segmentation performance of our approach on single or multiple object categories.

**Acknowledgments:** This work was supported in part by National Natural Science Foundation of China under Grant 62271431, in part by Research Grants Council of Hong Kong under Grants 25207822 & 15225522.

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

# Appendix

## A  DETAILS OF ARCHITECTURE AND DATE PREPARATION OF OBJECT-CENTRIC NETWORK

The object-centric network of our GrabS framework comprises an orientation estimation network, a Variational Autoencoder (VAE), and a Diffusion Network. These components share a common encoder architecture, which is based on PointNet++ followed by self-attention blocks. As depicted in Figure 9 (a), the encoder consists of four Set Abstraction (SA) blocks, each followed by a self-attention mechanism. The SA blocks are characterized by $K$ local regions with a ball radius $r$, followed by three Multi-Layer Perceptron (MLP) layers.

After extracting shape features from the encoder, the orientation estimation network employs a single MLP layer with 128 hidden neurons to regress the three rotation angles. The VAE utilizes one layer to output the mean and variance of a Gaussian distribution, from which a sample feature is drawn. The VAE's decoder comprises ten MLP layers, with input, output, and hidden dimensions of 259, 1, and 256, respectively, to regress the Signed Distance Function (SDF) for query points.

For the Diffusion model, the latent feature is obtained from the well-trained VAE, and the condition is embedded from the input shape using the encoder architecture shown in Figure 9(a). The denoising network is a three-layer MLP with dimensions of $\{256 \times 3 - 256 \times 2 - 256\}$. The input to the denoising MLP is the concatenation of the condition, noisy feature, and time embedding. The timestamp is normalized to the range $[0, 1]$ and its embedding is computed using sine and cosine functions.

For the data preparation in training the object-centric network, we utilize the same data for orientation estimation, VAE, and Diffusion models. During the segmentation of real-world scenes, the training data for the object-centric network is derived from the ShapeNet (Chang et al., 2015) chairs with occlusions provided by EFEM (Lai et al., 2023). These occlusions are generated by projected depth images. Additionally, we follow them to randomly incorporate ground, walls, and fragments of other objects to simulate real-world background conditions.

For synthetic scenes, we employ six classes from ShapeNet without introducing occlusions while still retaining the background data augmentation to mimic scene environments.

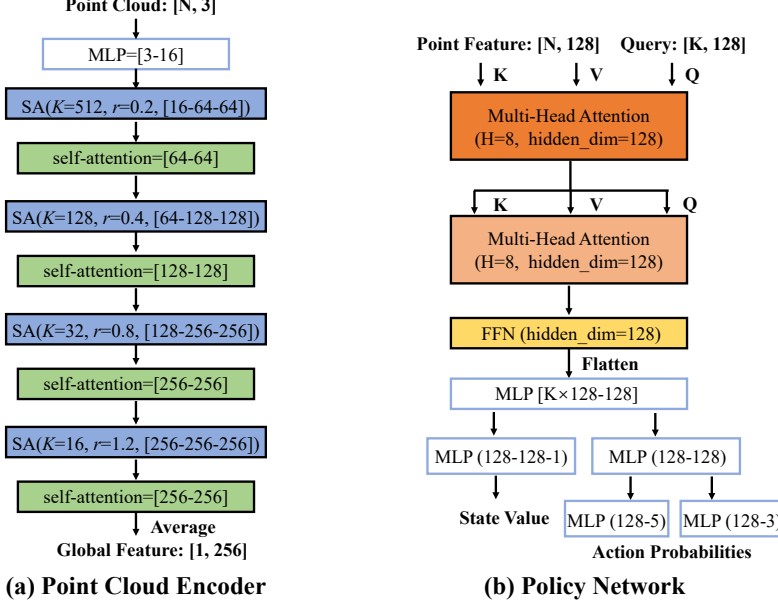

Figure 9: Details of (a) Point Cloud encoder used in orientation estimation network, VAE, and Diffusion; (b) Policy Newtork.

## B    ABLATION STUDY ON ATTENTION BLOCKS IN GRABS

Global information is important for learning the interior/exterior of the surface, as discussed in (Li et al., 2024; 2023b). Following their approach, we incorporate attention blocks in the encoder of our object-centric network. In this section, we conduct an ablation study on them. Table 7 shows the necessity of attention blocks.

Table 7: Ablation study of attention blocks in the object-centric network.

|  | AP(%) | AP50(%) | AP25(%) |
|---|---|---|---|
| Remove attention blocks | 42.9 | 66.3 | 76.9 |
| **Full GrabS** | **46.7** | **71.5** | **82.9** |

## C    ABLATION STUDY OF THE NUMBER OF PARALLEL TRAJECTORIES

We further conduct an ablation study about the number of trajectories created in parallel for object discovery branch. Here we choose 25/50/75/100 trajectories respectively, whereas we choose 50 in our main experiments.

Table 8 shows the results. We can see that, 1) the number of trajectories is not crucial once it is more than a certain number, *e.g.*, 50. 2) Too few trajectories can lead to a slight decrease in the final performance due to an insufficient number of object masks discovered.

Table 8: Ablation results on ScanNet validation set for different numbers of parallel trajectories.

| no. of trajectories | AP(%) | AP50(%) | AP25(%) |
|---|---|---|---|
| 25 | 42.0 | 64.1 | 74.4 |
| 50 | 46.7 | 71.5 | 82.9 |
| 75 | 46.9 | 69.5 | 80.8 |
| 100 | 47.1 | 69.7 | 81.3 |

## D    ABLATION STUDY ON DIFFERENT TYPES OF SUPERPOINTS

When training Mask3D on ScanNet, it uses the superpoints provided by ScanNet in the cross-attention block to group voxel features into superpoint features. These superpoint features then interact with query features through cross-attention. This process primarily aims to reduce the computational load and GPU memory usage, enhancing training and inference efficiency.

We further conduct experiments to assess the impact of superpoints provided by ScanNet dataset. In particular, we choose to use the following two new strategies: 1) using superpoints generated by SPG (Landrieu & Simonovsky, 2018) in an unsupervised manner, and 2) directly extracting features on voxels instead of superpoints.

Table 9 shows results on the validation set of ScanNet. We can see that directly using voxels without any superpoints can achieve comparable performance with that of ScanNet superpoints, though the latter is slightly better.

Table 9: Ablation results of different types of superpoints on the validation set of ScanNet.

|  | AP(%) | AP50(%) | AP25(%) |
|---|---|---|---|
| ScanNet superpoints | **46.7** | **71.5** | **82.9** |
| SPG superpoints | 43.7 | 61.9 | 69.1 |
| without superpoints | 45.1 | 65.2 | 72.3 |

# E  SENSITIVITY OF $\delta_c$ ON DIFFERENT DATASETS.

We further conduct a comprehensive ablation study on ScanNet and S3DIS to assess the sensitivity of the parameter $\delta_c$. From Table 10, we can see that $\delta_c$ should be typically set as 0.14 or 0.16 on two datasets. Nevertheless, since S3DIS has more occluded or distorted shapes than ScanNet, it is more challenging for our object-centric network to identify 3D objects in S3DIS. Therefore, $\delta_c$ is slightly larger (more relaxed) in S3DIS than in ScanNet.

Table 10: Ablation results of different $\delta_c$ on ScanNet validation set and S3DIS Area5.

|  |  | AP(%) | AP50(%) | AP25(%) |
|---|---|---|---|---|
| ScanNet | $\delta_c = 0.12$ | 42.6 | 60.8 | 67.6 |
|  | $\delta_c = 0.14$ | **46.7** | **71.5** | 82.9 |
|  | $\delta_c = 0.16$ | 43.8 | 68.8 | 81.0 |
|  | $\delta_c = 0.18$ | 43.8 | 70.1 | **85.0** |
|  | $\delta_c = 0.20$ | 42.5 | 69.9 | 84.2 |
| S3DIS Area5 | $\delta_c = 0.12$ | 46.2 | 65.7 | 71.7 |
|  | $\delta_c = 0.14$ | 46.4 | 66.2 | 73.8 |
|  | $\delta_c = 0.16$ | **51.3** | **81.8** | 86.0 |
|  | $\delta_c = 0.18$ | 48.3 | 83.1 | **90.5** |
|  | $\delta_c = 0.20$ | 44.9 | 79.1 | 88.7 |

# F  VISUALIZATIONS OF FAILURE CASES

The failure cases in our GrabS mainly include two types as shown in Figure 10. The first type is that our model mistakenly segments objects whose shapes are similar to the target shape (*e.g.*, chairs). For instance, it may incorrectly segment parts of a wall with a plane as a chair. The second type is missing some occluded chairs, primarily because those severely occluded chairs are hard to be reconstructed by the object-centric network.

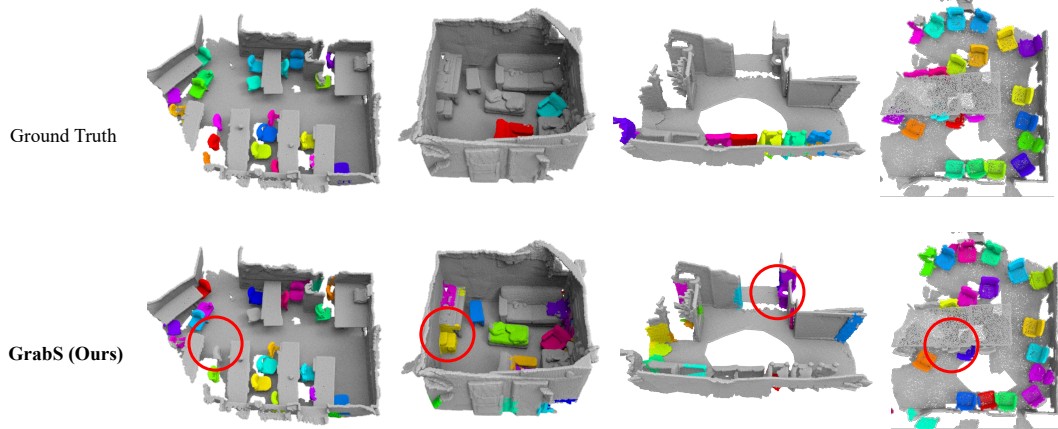

Figure 10: Failure cases of our method on ScanNet validation set.

## G  THE INFLUENCE OF DISCOVERED MASKS ON POLICY NETWORK LEARNING

To further explore the influence of previously discovered masks on the training of policy networks, we analyze the quality and accuracy of discovered candidate masks from the object discovery branch (embodied agent) at various epochs. In particular, we conduct the analysis on the training set of ScanNet. A discovered mask is considered as accurate if it has an IoU greater than 50% with any ground truth mask. We track the number of newly discovered masks, defined as those have never been identified in all previous epochs.

Table 11 shows the number and accuracy of discovered masks over a certain number of training epochs. We can see that: 1) Over training, the total number of discovered masks increases, and the accuracy improves. However, the number of newly discovered masks decreases over time, suggesting that the network becomes more consistent in identifying relevant masks as training advances. 2) The accuracy of newly discovered masks declines, primarily because these objects that are easier to reconstruct can be identified in early epochs. As training progresses, the model likely attempts to discover objects that are harder to represent, which is risky and error-prone.

Table 11: The number and accuracy of discovered objects over a certain number of epochs. Note that, some newly discovered masks may overlap with each other. All the numbers in this table are the counts after removing overlapping masks.

| Epoch | 0 | 50 | 100 | 150 | 200 | 250 | 300 | 350 | 400 |
|---|---|---|---|---|---|---|---|---|---|
| Mask Number | 920 | 5630 | 6981 | 7421 | 7457 | 7696 | 8133 | 7922 | 7931 |
| Mask Accuracy (%) | 16.5 | 36.2 | 41.0 | 43.5 | 42.0 | 42.2 | 44.0 | 44.7 | 45.5 |
| New Mask Number | 920 | 5257 | 4336 | 2404 | 1982 | 1716 | 1576 | 1277 | 870 |
| New Mask Accuracy (%) | 16.5 | 26.7 | 15.3 | 10.2 | 7.4 | 6.0 | 5.1 | 4.3 | 4.3 |

## H  CLASS-AGNOSTIC EVALUATION ON SCANNET AND S3DIS

Following Unscene3D and Part2Object, we also evaluate class-agnostic object segmentation on ScanNet and S3DIS. In particular, we directly use our object-centric network trained on the six classes of ShapeNet in Section 4.3, and then train our multi-object estimation network on the training set of ScanNet. Lastly, our object detection model is evaluated on the validation set of ScanNet, and also Area 5 of S3DIS as a cross-dataset evaluation. All baselines are trained and tested using the same datasets to ensure fairness in evaluation.

Table 12 shows the quantitative results. Note that, the baseline Part2Object (2D only) uses pretrained DINOv2 (Oquab et al., 2024) to provide object priors, Unscene3D (2D+3D) uses both DINO (Caron et al., 2021) and CSC (Hou et al., 2021) to provide object priors, and Unscene3D (3D only) uses CSC only to provide object priors. We can see that our model outperforms Unscene3D (3D only) but falls short of Unscene3D (2D+3D) and Part2Object (2D only), primarily because they leverage extremely rich object priors from large 2D models like DINO and DINOv2, whereas we only use object priors from limited six classes of ShapeNet dataset. We leave the use of larger scale 2D or 3D priors as our future exploration.

Table 12: Class agnostic segmentation results on ScanNet validation set and S3DIS Area5.

| | | AP(%) | AP50(%) | AP25(%) |
|---|---|---|---|---|
| | Part2Object (Shi et al., 2024) (2D only) | **16.9** | **36.0** | **64.9** |
| | Unscene3D (Rozenberszki et al., 2024) (2D+3D) | 15.9 | 32.2 | 58.5 |
| ScanNet | Unscene3D (Rozenberszki et al., 2024) (3D only) | 13.3 | 26.2 | 52.7 |
| | EFEM (Lei et al., 2023) | 6.4 | 13.7 | 21.5 |
| | GrabS(Ours-VAE) | 14.3 | 27.2 | 41.4 |
| | Part2Object (Shi et al., 2024) (2D only) | **8.7** | **19.4** | **40.8** |
| | Unscene3D (Rozenberszki et al., 2024) (2D+3D) | 8.5 | 16.7 | 35.5 |
| S3DIS Area5 | Unscene3D (Rozenberszki et al., 2024) (3D only) | 8.3 | 15.3 | 32.2 |
| | EFEM (Lei et al., 2023) | 4.6 | 7.3 | 12.3 |
| | GrabS(Ours-VAE) | 8.5 | 13.2 | 20.5 |

## I VISUALIZATIONS OF AGENT TRAJECTORIES

Figure 11 shows three trajectories of the agent, along with the candidate mask and the recovered full shape within each container.

## J MEMORY AND TIME COSTS

The first stage of our pipeline involves training an orientation estimator and a generative model. It takes 8 and 21 hours respectively, with GPU memory of 4GB and 8GB. The second stage takes 62 hours and GPU memory of 20GB to train the whole network.

Although our training process is more time-consuming than baselines, our inference speed and memory cost are the same as Unscene3D and Part2Object. It takes 0.092 seconds per scene and 5GB memory on average. This is significantly faster than EFEM which requires iterative inference with 2.3 minutes per scene and 8GB memory on average. The hardware for testing is a single RTX 3090 GPU with an AMD R9 5900X CPU.

## K DETAILS OF POLICY LEARNING

The policy network is designed to derive actions from input point features, functioning similarly to a target object detection mechanism within a block of point cloud. To achieve this, we emulate the segmentation head of Mask3D (Schult et al., 2023) in constructing our policy network. Specifically, as depicted in Figure 9 (b), we employ a transformer decoder to embed the information of target objects within the container, the initial query number is set as 32 in all our experimnets. Subsequently, we utilize three MLP layers to regress state values and an additional three MLP layers to predict actions.

We use Proximal Policy Optimization (PPO) algorithm to train the agent. The details of parameters in PPO, agent initialization, and reward computation are as follows.

For initialization, since the input to the policy network consists of points within a container, we initialize the cylinder container with a large radius, specifically $C_d = 2.0$m. This large radius provides the policy network with a substantial receptive field, ensuring it has sufficient information to determine subsequent actions.

Regarding the reward computation, if the points within the identified masks can be reconstructed, which is indicated by their Chamfer distance to the extracted meshes being less than the threshold $\delta_c$, we assign a reward score of 10 to this state and terminate the trajectory. Otherwise, a score of -1 is assigned. The maximum step length in our approach is set to be 8.

In PPO, we constrain the maximum change ratio between previous and current action distributions to 20% to ensure that policy distributions do not change too rapidly. We employ generalized advantage estimation rather than regressing the vanilla advantage. The balance parameter $\lambda$ is set as 0.5, and the discount weight of future return is set as 0.9. To encourage the exploration of actions, we apply an entropy loss to the action distribution. Therefore, there are three loss functions: the vanilla state value regression loss and PPO-Clip loss, together with an additional entropy loss whose coefficients are 1, 1, and 0.1. The optimizer is Adam with a learning rate of 0.0001 in all training epochs.

The parameters of PPO are set to be the same on all scene datasets. In implementation, we split the whole 3D scene into 50 blocks and initialize an agent in each block for parallel searching. The block size is set as 2.0m. We provide ablation experiments in Table 13 for the choice of block sizes.

## L DETAILS OF TRAINING SEGMENTATION BRANCH

To train the segmentation branch of our object estimation network. We basically follow Mask3D (Schult et al., 2023), while for more efficient training, we use a 5cm voxel size. The optimizer is AdamW with a learning rate of 0.0001 in all training epochs.

The Custom30M version of SparseConv (Choy et al., 2019) with a transformer decoder is chosen as the backbone and segmentation head. We simply use one transformer decoder block for efficient

training. In the transformer decoder, each initial query feature will be updated after the attention layers and then act as the center feature for each mask.

Mask3D incorporates three loss functions: binary cross-entropy and dice loss for mask supervision, and cross-entropy loss for mask classification. This classification is actually applied to the mask center features. we adopt these three loss functions directly. For the classification loss, we take the masks that can match with pseudo masks as foreground and others as background, so it is a binary classification loss in our setting. We take the original weighted combination of three losses as our segmentation loss, *i.e.*, 2/5/2. These loss functions and networks keep the same on all datasets.

Table 13: Results of ablated models of our GrabS on the ScanNet validation set. The bold settings are chosen in our full framework.

| | AP(%) | AP50(%) | AP25(%) | RC(%) | RC50(%) | RC25(%) | PR(%) | PR50(%) | PR25(%) |
|---|---|---|---|---|---|---|---|---|---|
| (1) block radius = 1.0m | 42.2 | 61.5 | 73.2 | 47.6 | 65.5 | 76.9 | 50.9 | 74.3 | 82.1 |
| **(2) block radius = 2.0m** | **46.7** | **71.5** | **82.9** | **53.2** | **74.5** | **85.2** | 52.1 | 76.4 | 83.0 |
| (2) block radius = 3.0m | 41.1 | 60.4 | 70.9 | 45.8 | 62.5 | 72.5 | 58.0 | 81.4 | 90.7 |

## M  DETAILS OF OUR SYNTHETIC DATASET

Following (Song & Yang, 2022), we generate 5000 static scenes with $4 \sim 8$ objects. The aspect ratio of the ground plane in each scene is uniformly sampled between 0.6 and 1.0. For each object in a scene, its scale is set as 1. Each object is scaled to be a unit size, and its rotation around the vertical z-axis is randomly sampled from $-180° \sim 180°$ . To simulate realistic indoor environments, walls and ground planes are created in the scenes. The resulting point clouds contain only coordinates without color information. Each scene has 20000 points.

To avoid object overlap, objects are placed sequentially within each scene. The bounding box of each newly placed object is checked against those of previously placed objects. If an overlap is detected, the object's position is adjusted until a non-overlapping location is found or the maximum number of placement attempts is reached. If a non-overlapping location cannot be found until the maximum attempt, we will drop this room. The maximum number of placement attempts is 1000.

## N  TRAINING AND EVALUATION ON SCANNET

We train our model on the ScanNet training set for 450 epochs with a batch size of 8. The number of queries in the transformer decoder is set as 50 in this dataset. We use the superpoints provide by ScanNet in training and inference. Figure 12 provides additional qualitative comparisons with baseline methods on the ScanNet validation set.

## O  EVALUATION ON S3DIS

Tabs. 14 to 19 show the results of cross dataset validation on each area of S3DIS. Figure 13 gives more qualitative comparisons.

## P  TRAINING AND EVALUATION ON OUR SYNTHETIC DATASET

Training hyperparameters are the same as used in ScanNet. The query number in Mask3D is set as 10 because there are up to 8 objects in each scene. We train our model for 150 epochs on this dataset with a batch size of 10. The superpoints are constructed by SPG. More visualizations are listed in Figure 14.

Table 14: Cross dataset evaluation of our method and baselines on the Area-1 of S3DIS.

| | | AP(%) | AP50(%) | AP25(%) | RC(%) | RC50(%) | RC25(%) | PR(%) | PR50(%) | PR25(%) |
|---|---|---|---|---|---|---|---|---|---|---|
| Unsupervised | Unscene3D (Rozenberszki et al., 2024) | 33.6 | 63.8 | 85.4 | 44.8 | 70.3 | 88.3 | 13.3 | 21.4 | 26.1 |
| | Part2Object (Shi et al., 2024) | 27.1 | 55.4 | 77.5 | 38.8 | 66.4 | 83.2 | 37.8 | 65.2 | 80.6 |
| | EFEM (Lei et al., 2023) | 19.1 | 48.6 | 54.7 | 23.8 | 49.7 | 54.8 | 43.7 | 93.9 | **98.8** |
| | GrabS (Ours-VAE) | 45.5 | 68.6 | 73.2 | 51.3 | 71.0 | 75.5 | 62.7 | 88.0 | 90.7 |
| | **GrabS (Ours-Diffusion)** | **47.8** | **70.9** | **75.7** | **52.5** | **72.3** | **76.8** | **64.6** | **91.1** | 92.2 |

Table 15: Cross dataset evaluation of our method and baselines on the Area-2 of S3DIS.

| | | AP(%) | AP50(%) | AP25(%) | RC(%) | RC50(%) | RC25(%) | PR(%) | PR50(%) | PR25(%) |
|---|---|---|---|---|---|---|---|---|---|---|
| Unsupervised | Unscene3D (Rozenberszki et al., 2024) | 3.2 | 6.0 | 10.5 | 5.0 | 8.0 | 13.2 | 7.0 | 11.5 | 18.4 |
| | Part2Object (Shi et al., 2024) | 2.8 | 6.2 | 8.8 | 4.9 | 7.9 | 10.6 | 30.4 | 50.0 | 69.0 |
| | EFEM (Lei et al., 2023) | 1.1 | 2.9 | 9.2 | 1.6 | 3.5 | 9.2 | 17.7 | 40.4 | **100.0** |
| | **GrabS (Ours-VAE)** | **6.4** | **10.2** | **17.2** | **8.0** | **11.7** | **18.5** | **35.2** | **53.8** | 78.9 |
| | GrabS (Ours-Diffusion) | 5.3 | 8.1 | 13.3 | 6.2 | 8.8 | 14.5 | 28.9 | 43.6 | 67.5 |

Table 16: Cross dataset evaluation of our method and baselines on the Area-3 of S3DIS.

| | | AP(%) | AP50(%) | AP25(%) | RC(%) | RC50(%) | RC25(%) | PR(%) | PR50(%) | PR25(%) |
|---|---|---|---|---|---|---|---|---|---|---|
| Unsupervised | Unscene3D (Rozenberszki et al., 2024) | 37.6 | 58.2 | **83.6** | 51.0 | 68.6 | **88.0** | 16.2 | 22.3 | 26.4 |
| | Part2Object (Shi et al., 2024) | 38.9 | 63.5 | 81.4 | 49.9 | 73.1 | 86.5 | 43.6 | 65.3 | 78.4 |
| | EFEM (Lei et al., 2023) | 29.0 | 58.3 | 64.2 | 35.8 | 59.7 | 64.2 | 55.4 | 95.2 | **100.0** |
| | **GrabS (Ours-VAE)** | **59.5** | **78.8** | 80.2 | **62.9** | **79.1** | 80.6 | **73.4** | 93.0 | 94.7 |
| | GrabS (Ours-Diffusion) | 51.4 | 67.0 | 67.0 | 52.7 | 67.2 | 67.2 | 76.7 | 97.8 | 97.8 |

Table 17: Cross dataset evaluation of our method and baselines on the Area-4 of S3DIS.

| | | AP(%) | AP50(%) | AP25(%) | RC(%) | RC50(%) | RC25(%) | PR(%) | PR50(%) | PR25(%) |
|---|---|---|---|---|---|---|---|---|---|---|
| Unsupervised | Unscene3D (Rozenberszki et al., 2024) | 23.9 | 49.1 | 70.8 | 35.8 | 60.3 | 76.7 | 11.6 | 19.9 | 24.9 |
| | Part2Object (Shi et al., 2024) | 26.8 | 57.4 | 75.3 | 38.7 | 67.3 | **79.8** | 38.2 | 66.5 | 80.3 |
| | EFEM (Lei et al., 2023) | 15.1 | 36.5 | 49.1 | 19.9 | 37.7 | 49.1 | 42.7 | 90.9 | **100.0** |
| | GrabS (Ours-VAE) | 39.2 | **66.4** | **73.4** | 45.1 | **67.9** | 74.8 | 53.9 | 81.8 | 86.2 |
| | **GrabS (Ours-Diffusion)** | **39.4** | 64.8 | 68.0 | **46.1** | 66.0 | 69.2 | **63.9** | **92.9** | 94.8 |

Table 18: Cross dataset evaluation of our method and baselines on the Area-5 of S3DIS.

| | | AP(%) | AP50(%) | AP25(%) | RC(%) | RC50(%) | RC25(%) | PR(%) | PR50(%) | PR25(%) |
|---|---|---|---|---|---|---|---|---|---|---|
| Unsupervised | Unscene3D (Rozenberszki et al., 2024) | 42.6 | 63.4 | **80.3** | 51.9 | 68.6 | **83.3** | 17.4 | 23.5 | 27.7 |
| | Part2Object (Shi et al., 2024) | 30.0 | 50.5 | 76.4 | 45.2 | 64.7 | 82.6 | 43.5 | 62.5 | 80.1 |
| | EFEM (Lei et al., 2023) | 14.9 | 35.7 | 45.3 | 18.6 | 36.0 | 45.3 | 43.6 | **92.1** | **100.0** |
| | **GrabS (Ours-VAE)** | **46.4** | **66.2** | 73.8 | 51.0 | 67.1 | 74.0 | 68.5 | 91.5 | 97.0 |
| | GrabS (Ours-Diffusion) | 44.2 | 58.0 | 62.6 | 45.7 | 58.9 | 63.2 | **70.8** | 91.6 | 96.4 |

Table 19: Cross dataset evaluation of our method and baselines on the Area-6 of S3DIS.

| | | AP(%) | AP50(%) | AP25(%) | RC(%) | RC50(%) | RC25(%) | PR(%) | PR50(%) | PR25(%) |
|---|---|---|---|---|---|---|---|---|---|---|
| Unsupervised | Unscene3D (Rozenberszki et al., 2024) | 41.3 | 70.9 | **92.3** | 51.6 | 75.9 | **94.4** | 12.8 | 19.6 | 23.6 |
| | Part2Object (Shi et al., 2024) | 26.5 | 57.4 | 83.0 | 43.0 | 74.9 | 91.1 | 33.6 | 58.8 | 71.5 |
| | EFEM (Lei et al., 2023) | 18.3 | 45.2 | 53.1 | 23.6 | 45.8 | 53.1 | 46.1 | 94.3 | **99.0** |
| | **GrabS (Ours-VAE)** | **52.1** | **80.4** | **84.3** | **57.3** | **80.4** | 84.4 | **66.6** | 96.0 | 96.8 |
| | GrabS (Ours-Diffusion) | 47.1 | 74.5 | 76.2 | 52.2 | 76.0 | 77.7 | 65.6 | 96.5 | 97.9 |

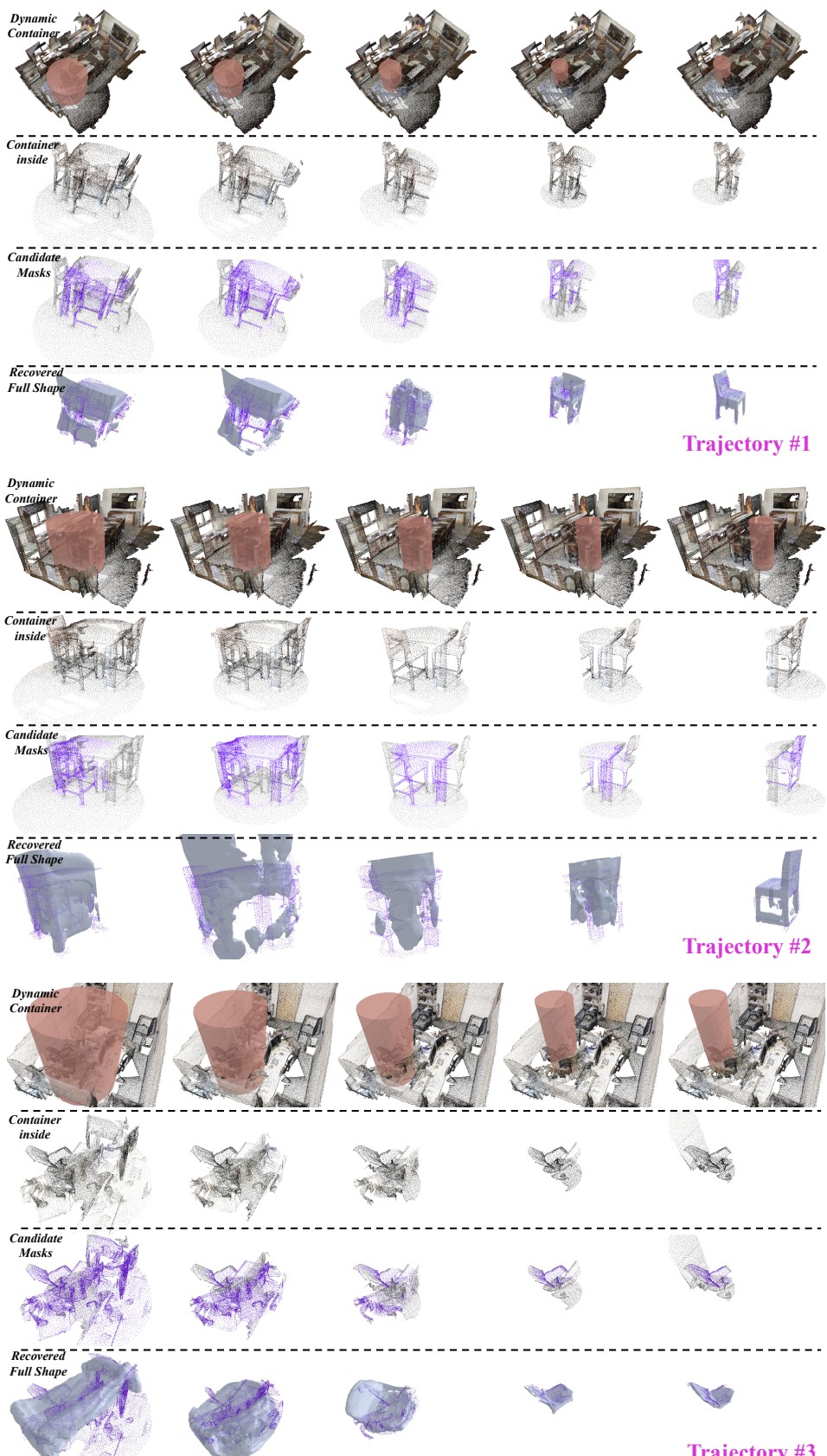

Figure 11: Sample trajectories of the agent.

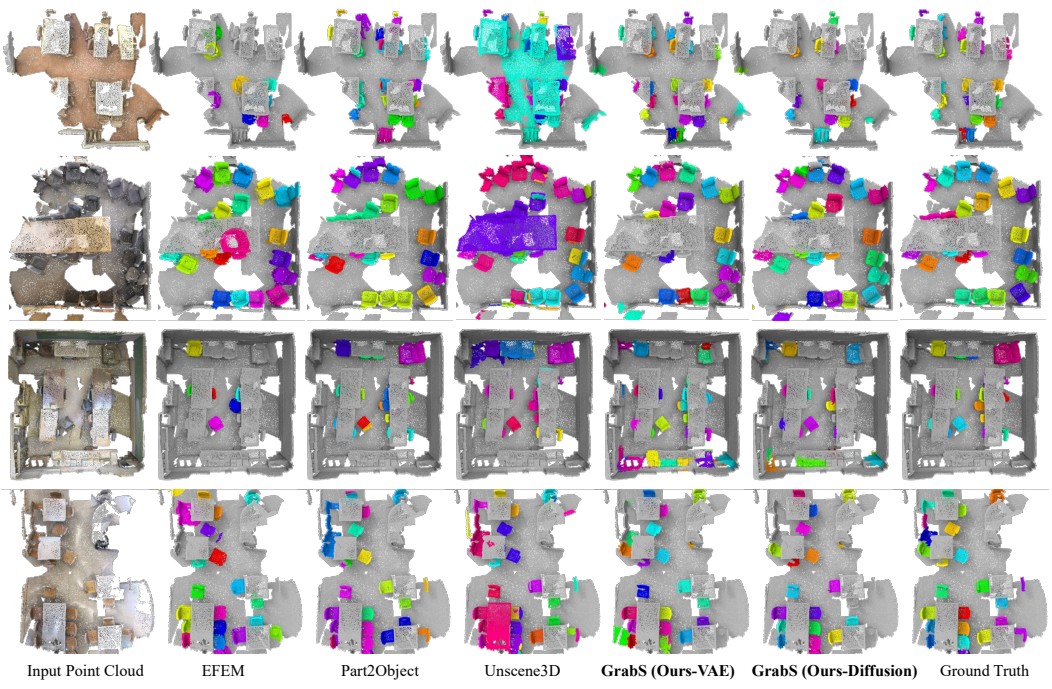

Figure 12: More qualitive results on ScanNet validation set.

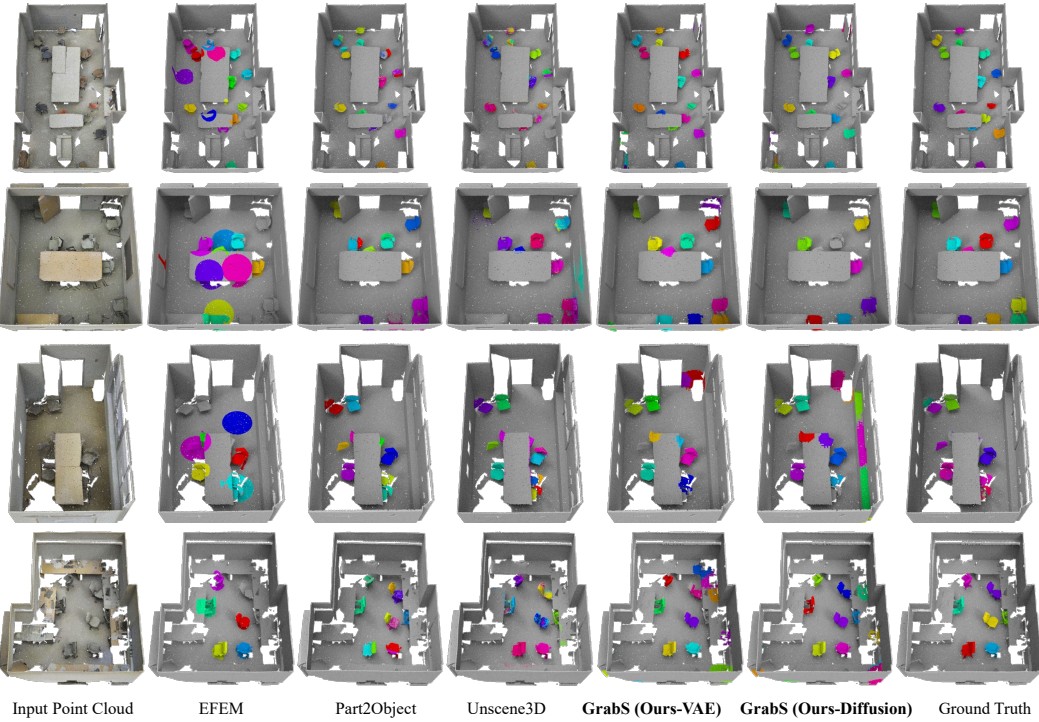

Figure 13: More qualitive results on S3DIS.

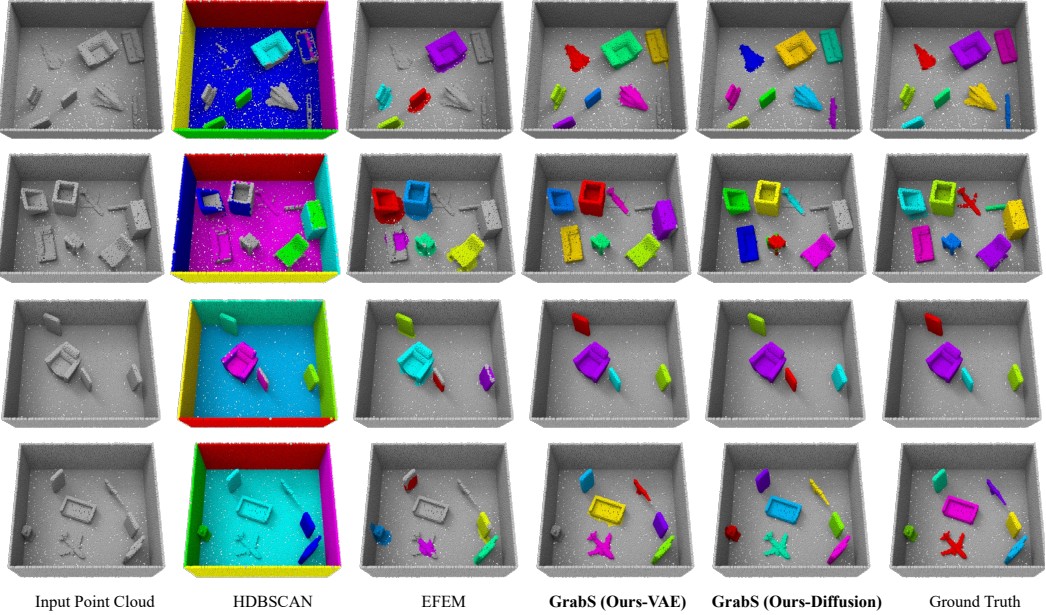

Figure 14: More qualitive results on the test set of our synthetic dataset.

