# OpenReview forum: "GrabS: Generative Embodied Agent for 3D Object Segmentation without Scene Supervision"
_ICLR.cc/2025/Conference — ICLR 2025 Spotlight_

### Official Review · Reviewer_Bja3 · 2024-11-03

**Soundness:** 3
**Presentation:** 3
**Contribution:** 3
**Rating:** 6
**Confidence:** 3

**Summary:**

To overcome the limitations of prior works, which require labor-intensive large-scale annotations, the authors explore the challenging problem of 3D instance segmentation in complex point clouds without relying on human labels.

They introduce a two-stage unsupervised 3D instance segmentation framework, GOPS: (1) in the first stage, an object-centric network is trained to learn generative object-centric priors, and (2) in the second stage, a multi-object estimation network then identifies similar 3D objects by querying against the learned priors and receiving objectiveness rewards through a reinforcement learning strategy.

Extensive experiments on two real-world datasets, ScanNet and S3DIS, and a newly created synthetic dataset demonstrate the effectiveness of GOPS with superior 3D instance segmentation performance.

**Strengths:**

- The paper is well-organized, with clear explanations in both text and diagrams for each section.

- Based on a thorough analysis of prior methods and their limitations, the authors clearly articulate the motivation underlying their proposed approach.

- Each module in the two-stage pipeline is technically sound to improve the unsupervised 3D instance segmentation without relying on large-scale human labels.

**Weaknesses:**

- The experiments for each module are somewhat lacking.
In addition to the points mentioned below, it would be helpful to provide experiments that validate the detailed performance of each module.
1) If the agent is indeed well-trained, it would be better to visualize the regions discovered by the agent or trajectories of the agent during exploration.
2) While the ultimate goal of GOPS is instance segmentation, the performance of the object-centric network seems to be of significant importance.
Therefore, providing relevant experimental results for the object-centric network would further solidify the effectiveness of the network.
For example, providing visualizations of the input point cloud of the trained object-centric network along with the corresponding recovered full shape would be beneficial.

- The authors conducted training and evaluation solely on the chair class of real-world datasets (ScanNet and S3DIS).
While they evaluate performance on the synthetic dataset with six classes, the model's effectiveness in real-world scenarios, including various instance categories, remains unclear.
Is it possible to train and evaluate GOPS for six class objects from real-world datasets (ScanNet and S3DIS) using the object-centric network trained for the six class objects used in the synthetic dataset experiments?
Or train object-centric networks for six class objects in real-world datasets again?

- While GOPS does not require labor-intensive human annotations, training the two-stage GOPS frameworks seems to demand heavy resources.
It would be helpful to clarify their implementation details, including the memory and time costs for the training process.

**Questions:**

- Can the proposed object-centric network learn knowledge about multiple objects simultaneously?
In the experiments on the synthetic dataset, did the authors train separate object-centric networks for each of the six objects or utilize a single object-centric network that learns from all six objects?

- Could the authors explain the rationale behind adding a self-attention block to the encoder in the object-centric network?

---

> ### Author Response · Authors · 2024-11-26
> **Responses to Reviewer Bja3 (Part1)**
>
> We appreciate the reviewer’s thoughtful comments and address the main concerns below.
>
> ***Q1: The experiments for each module are somewhat lacking. In addition to the points mentioned below, it would be helpful to provide experiments that validate the detailed performance of each module.***
>
> **A1**: This is a very valuable suggestion. As requested, we further conduct the following two more ablation experiments regarding each component of our multi-object estimation network. In particular,
>
> - ****Removing the Object Discovery Branch****: This is to evaluate the effectiveness of using RL to aid object discovery. Without it, we randomly select 50 positions in each scene and randomly set a radius ranging from $0\sim 2$ meters as the container for discovering objects as pseudo labels.
>
> - ****Removing the Object Segmentation Branch****: This is to evaluate the effectiveness of the Object Segmentation Branch. Without it, we only use the Object Discovery Branch to collect object masks by querying against the frozen object-centric network.
>
> The following Table 1 shows our new ablation results. We also include our existing ablation results on the two components of our object-centric network. We can see that each component of our pipeline plays its role, and together they enable our full method to achieve excellent performance.
>
> *Table1: Ablation studies on each module in GOPS.*
> |                | AP (%) | AP50 (%) | AP25 (%) |
> |--------------------------------------|--------|----------|----------|
> | Replace VAE by AE                    | 32.0   | 57.1     | 76.7     |
> | Remove orientation estimator         | 35.3   | 56.3     | 69.7     |
> | Remove Object Discovery Branch       | 34.2   | 56.7     | 69.4     |
> | Remove Object Segmentation Branch    | 25.7   | 47.9     | 55.3     |
> | Full GOPS(VAE)                       | 46.7   | 71.5     | 82.9     |
>
> In the revised paper, we have added the new ablations and results in Section 4.4.
>
> ***Q2: If the agent is indeed well-trained, it would be better to visualize the regions discovered by the agent or trajectories of the agent during exploration.***
>
> **A2**: Thanks for the advice. In Figure 12 of Appendix J in the revised paper, we have provided sample trajectories of the agent. In each step of a trajectory, we also visualize the discovered candidate mask and the recovered full shape within each container. Visualization of trajectories has also been provided in our originally submitted video demo in the Supplementary Material.
>
> ***Q3: While the ultimate goal of GOPS is instance segmentation, the performance of the object-centric network seems to be of significant importance. Therefore, providing relevant experimental results for the object-centric network would further solidify the effectiveness of the network. For example, providing visualizations of the input point cloud of the trained object-centric network along with the corresponding recovered full shape would be beneficial.***
>
> **A3**: Very valuable suggestion. Our object-centric network aims to provide shape priors to the multi-object estimation network, so it is not required to be exceptionally skilled at full shape reconstruction. As requested, we further evaluate its performance. In particular, we compute the Chamfer distance on chair class in the val set of ShapeNet. Following ONet [1] and ConvOcc [2], we randomly sample 1024 points from a complete object point cloud or a partial object point cloud (converted from a single depth view) as input, and then randomly sample 100k points from both the recovered mesh and ground truth mesh to compute the Chamfer-L1 score.
>
> The following Table 2 shows the quantitative results and Figure 11 in the revised paper illustrates  qualitative results. Overall, our object-centric network achieves a similar reconstruction performance to ConvOcc.
>
> *Table2: Evaluation of our object-centric network for shape reconstruction.*
> |                | ONet             | ConvOcc   | Ours (complete)   | Ours (partial)   |
> |-------------|:-------------------:|:----------------:|:---------------------------:|:---------------------:|
> | Chamfer-L1 | 0.228              | 0.046  |     0.042  |     0.052

---

> > ### Author Response · Authors · 2024-11-26
> > **Responses to Reviewer Bja3 (Part2)**
> >
> > ***Q4: The authors conducted training and evaluation solely on the chair class of real-world datasets (ScanNet and S3DIS). While they evaluate performance on the synthetic dataset with six classes, the model's effectiveness in real-world scenarios, including various instance categories, remains unclear. Is it possible to train and evaluate GOPS for six class objects from real-world datasets (ScanNet and S3DIS) using the object-centric network trained for the six class objects used in the synthetic dataset experiments? Or train object-centric networks for six class objects in real-world datasets again?***
> >
> > **A4**: Following the reviewer's suggestion, we directly use our object-centric network trained on the six classes of ShapeNet in Section 4.3, and then train our multi-object estimation network on the training set of ScanNet. Lastly, our object detection model is evaluated on the validation set of ScanNet, and also Area 5 of S3DIS as a cross-dataset evaluation. Both baselines Unscene3D and Part2Object are trained and tested using the same datasets to ensure fairness for a comparison.
> >
> > The following Table 3 shows quantitative results. Note that, the baseline Part2Object (2D only) uses pre-trained DINOv2 [5] to provide object priors, Unscene3D (2D+3D) uses both DINO [6]]and CSC [7] to provide object priors, and Unscene3D (3D only) uses CSC only to provide object priors. We can see that our model outperforms Unscene3D (3D only) but falls short of Unscene3D (2D+3D) and Part2Object (2D only), primarily because they  leverage extremely rich object priors from large 2D models like DINO and DINOv2, whereas we only use object priors from limited six classes of ShapeNet dataset. We leave the use of larger scale 2D or 3D priors as our future exploration.
> >
> > *Table3: Class agnostic segmentation results on ScanNet validation set and S3DIS Area5.*
> > |               |                   | AP (%) | AP50 (%) | AP25 (%) |
> > |---------------|-------------------------|--------|----------|----------|
> > | **ScanNet**   | Part2Object (2D only)   | 16.9   | 36.0     | 64.9     |
> > |               | Unscene3D (2D+3D)       | 15.9   | 32.2     | 58.5     |
> > |               | Unscene3D (3D only)     | 13.3   | 26.2     | 52.7     |
> > |               | EFEM                    | 6.4    | 13.7     | 21.5     |
> > |               | GOPS(VAE)               | 14.3   | 27.2     | 41.4     |
> > | **S3DIS Area5** | Part2Object (2D only) | 8.7    | 19.4     | 40.8     |
> > |               | Unscene3D (2D+3D)       | 8.5    | 16.7     | 35.5     |
> > |               | Unscene3D (3D only)     | 8.3    | 15.3     | 32.2     |
> > |               | EFEM                    | 4.6    | 7.3      | 12.3     |
> > |               | GOPS(VAE)               | 8.5    | 13.2     | 20.5     |
> >
> > In the revised paper, we have added the new results in Table 13 of Appendix I.
> >
> > ***Q5: While GOPS does not require labor-intensive human annotations, training the two-stage GOPS frameworks seems to demand heavy resources. It would be helpful to clarify their implementation details, including the memory and time costs for the training process.***
> >
> > **A5**: The first stage of our pipeline involves training an orientation estimator and a generative model. It takes 8 and 21 hours respectively, with GPU memory of 4GB and 8GB. The second stage takes 62 hours and GPU memory of 20GB  to train the whole network.
> >
> > Although our training process is more time-consuming than baselines, our inference speed and memory cost are the same as Unscene3D and Part2Object. It takes 0.092 seconds per scene and 5GB memory on average. This is significantly faster than EFEM which requires iterative inference with 2.3 minutes per scene and 8GB memory on average. The hardware for testing is a single RTX 3090 GPU with an AMD R9 5900X CPU.
> >
> > We will release our code in the next version.
> >
> > ***Q6: Can the proposed object-centric network learn knowledge about multiple objects simultaneously? In the experiments on the synthetic dataset, did the authors train separate object-centric networks for each of the six objects or utilize a single object-centric network that learns from all six objects?***
> >
> > **A6**: We train a single object-centric network on all six categories, enabling it to learn priors about multiple types of objects.
> >
> > In the revised paper, we have clarified this point in line 422 of Section 4.3.

---

> ### Author Response · Authors · 2024-11-26
> **Responses to Reviewer Bja3 (Part3)**
>
> ***Q7: Could the authors explain the rationale behind adding a self-attention block to the encoder in the object-centric network?***
>
> **A7**: We appreciate this detailed question. As discussed in prior works [3, 4], determining the interior/exterior of a surface reliably requires more global information than a local patch. Therefore, we follow them to use attention blocks to capture global geometry. To validate this point, we further conduct an ablation study on the ScanNet validation set by removing the attention blocks. We also compute Chmafer-L1 scores (calculation steps refer to A3) to evaluate the reconstruction performance without attention blocks .
>
> The following Tables 4 and 5 show the results. We can see that the incorporation of attention blocks is indeed beneficial for our object-centric network to learn better object priors.
>
> *Table4: Ablation study of attention blocks in the object-centric network.*
> |             | AP (%) | AP50 (%) | AP25 (%) |
> |--------------------------|--------|----------|----------|
> | Remove attention blocks  | 42.9   | 66.3     | 76.9     |
> | Full GOPS                | 46.7   | 71.5     | 82.9     |
>
> *Table5: Reconstruction performance of the object-centric network.*
> |            | ONet  | ConvOcc | Ours (complete) | Ours (complete)| Ours (partial) | Ours (partial)                |
> |------------|:-------:|:---------:|:----------------:|:-----------------:|:--------------:|:-----------------:|
> |            |       |         | w/ att   | w/o att | w/ att | w/o att |
> | Chamfer-L1 | 0.228 | 0.046   | 0.042            | 0.047           | 0.052          | 0.060           |
>
> In the revised paper, we have added these new results in Table 7 of Appendix B.
>
> **References:**
>
> [1]. Lars Mescheder, Michael Oechsle, Michael Niemeyer, Sebastian Nowozin, and Andreas Geiger.
> Occupancy networks: Learning 3d reconstruction in function space. CVPR, 2019.
>
> [2]. Songyou Peng, Michael Niemeyer, Lars Mescheder, Marc Pollefeys, and Andreas Geiger. Convolu-
> tional occupancy networks. ECCV, 2020.
>
> [3]. Qing Li, Huifang Feng, Kanle Shi, Yue Gao, Yi Fang, Yu-Shen Liu, and Zhizhong Han. Learning
> signed hyper surfaces for oriented point cloud normal estimation. IEEE TPAMI, 2024
>
> [4]. Qing Li, Huifang Feng, Kanle Shi, Yue Gao, Yi Fang, Yu-Shen Liu, and Zhizhong Han. Shs-net:
> Learning signed hyper surfaces for oriented normal estimation of point clouds. CVPR, 2023
>
> [5]. Maxime Oquab, Timoth´ee Darcet, Th´eo Moutakanni, Huy V Vo, Marc Szafraniec, Vasil Khali-
> dov, Pierre Fernandez, Daniel Haziza, Francisco Massa, Alaaeldin El-nouby, Mahmoud Assran,
> Nicolas Ballas, Wojciech Galuba, Russell Howes, Po-yao Huang, Shang-wen Li, Ishan Misra,
> Michael Rabbat, Vasu Sharma, Gabriel Synnaeve, Hu Xu, Herv´e Jegou, and Julien Mairal. DI-
> NOv2: Learning Robust Visual Features without Supervision. TMLR, 2024
>
> [6]. Mathilde Caron, Hugo Touvron, Ishan Misra, Herv´e J´egou, Julien Mairal, Piotr Bojanowski, and
> Armand Joulin. Emerging Properties in Self-Supervised Vision Transformers. ICCV, 2021
>
> [7]. Ji Hou, Benjamin Graham, Matthias Nießner, and Saining Xie. Exploring data-efficient 3d scene
> understanding with contrastive scene contexts. CVPR, 2021.

---

> > ### Comment · Reviewer_Bja3 · 2024-11-28
> >
> > Thanks to the authors for their thorough responses and extensive additional experiments!
> > Through these experimental discussions, most of my concerns have been addressed.
> > Therefore, I will maintain my positive rating.
> > Regarding the results of Q4, I understand that the lower class-agnostic scores are due to the lack of prior information.
> > However, providing scores for the two objects (sofa and cabinet), which are included in the labels of ScanNet and S3DIS, would further demonstrate the applicability of the proposed method.

---

> ### Author Response · Authors · 2024-11-30
> **Responses to Comments from Reviewer Bja3**
>
> **Comment: Regarding the results of Q4, I understand that the lower class-agnostic scores are due to the lack of prior information. However, providing scores for the two objects (sofa and cabinet), which are included in the labels of ScanNet and S3DIS, would further demonstrate the applicability of the proposed method.**
>
> **Response:** As requested, we evaluate our model from Q4 on the *sofa* and *cabinet* classes. Since our model and all baselines employ class-agnostic segmentation, we assign a ground truth class label to each mask predicted by our model and baselines, only selecting the masks corresponding to *sofa* and *cabinet* classes for evaluation on the ScanNet val set and S3DIS Area 5. Since S3DIS does not have the *cabinet* class, we only evaluate the *sofa* class here. As shown in the following Tables 6/7, our model generally outperforms all baselines on the two classes. Since our object-centric network is trained on six categories, including *sofa*, and the multi-object estimation network is trained on the ScanNet train set, our results for *sofa* on the ScanNet val set are rather satisfactory. We observe that the results of all methods for *sofa* on S3DIS Area 5 are notably low, primarily because the specific category *sofa* is extremely challenging to segment on the S3DIS dataset, even for existing fully-supervised methods. These new results will be added to our paper in the next version.
>
> *Table 6: Segmentation results of *sofa* and *cabinet* on ScanNet validation set.*
> | Class      | Method                          | AP (%) | AP50 (%) | AP25 (%) |
> |------------|---------------------------------|--------|----------|----------|
> | **sofa**   | Part2Object (2D only)           | 15.5   | 33.6     | 66.0     |
> |            | Unscene3D (2D+3D)               | 34.5   | 79.2     | 93.3     |
> |            | Unscene3D (3D only)             | 26.8   | 52.2     | 83.8     |
> |            | EFEM                            | 17.6   | 33.8     | 55.1     |
> |            | GOPS (VAE)                      | 36.8   | 59.3     | 72.0     |
> | **cabinet**| Part2Object (2D only)           | 8.1    | 23.5     | 51.6     |
> |            | Unscene3D (2D+3D)               | 4.7    | 16.1     | 45.0     |
> |            | Unscene3D (3D only)             | 3.4    | 12.2     | 41.2     |
> |            | EFEM                            | 1.7    | 4.8      | 42.5     |
> |            | GOPS (VAE)                      | 8.5    | 20.7     | 52.1     |
>
> *Table 7: Segmentation results of *sofa* on S3DIS Area 5.*
> | Class      | Method                          | AP (%) | AP50 (%) | AP25 (%) |
> |------------|---------------------------------|--------|----------|----------|
> | **sofa**   | Part2Object (2D only)           | 2.9    | 12.8     | 32.8     |
> |            | Unscene3D (2D+3D)               | 10.5   | 20.8     | 52.2     |
> |            | Unscene3D (3D only)             | 6.4    | 29.8     | 42.4     |
> |            | EFEM                            | 5.1    | 9.1      | 45.5     |
> |            | GOPS (VAE)                      | 11.1   | 27.3     | 33.3     |

---

### Official Review · Reviewer_xEK4 · 2024-11-03

**Soundness:** 4
**Presentation:** 3
**Contribution:** 3
**Rating:** 8
**Confidence:** 4

**Summary:**

The authors propose a pipeline with multiple components to identify instances in 3D scenes without human annotations. First, they train an object orientation module, followed by training a generative prior network that is tasked with recovering objects with different kinds of noises and obstructions. This network acts as a filter in a reinforcement based learning setting, where a cylinder is used to search the entire 3D point cloud for instances that match the patterns learned by the generative object prior network. Overall, the method works well and the authors conduct ablations on the methods components.

**Strengths:**

- The ideas presented in the paper are intuitive and make sense
- The writing is easy to follow and describes the contributions well
- The authors examine different contemporary learning mechanisms for their generative prior learning module, not just VAE but also diffusion
- First teaching a network what an instance should look like, the iteratively searching the 3D space with this sort of filter to identify instances, makes total sense, is quite intriguing and well executed
- The authors have conducted a good amount of ablations to observe different aspects of their method

**Weaknesses:**

- The paper investigates the sensibility of the threshold δc on 1 dataset, which is fine, but it would be interesting to know if this threshold is general or needs to be tuned individually for each dataset
- It would be interesting to know how a successful discovery of a mask influences the next iteration of the policy network
- In Figure 5, it says the scene in cropped and then encoded, while in section 3.3, the authors seem to argue against random cropping. In the beginning, the container-based cropping should also be close to random right? Is the idea here that the cropping will become more targeted as the container is better navigated by the policy network? Please clear up the confusion
- What puzzles me is how the qualitative results with diffusion prior look better in some cases that the VAE based ones (Fig 6 row 1, Fig 7 row 3), but this is not reflected in the quantitative evaluation. Could you provide an intuition for this is the case? Can you also show failure cases?
- It would be nice if acronyms like SDF would be introduced. Even though this is an established method, its still also done for acronyms like ViT=Vision Transformer.

**Questions:**

Overall, I think the idea of the paper is quite neat! The writing is well executed, the results well presented and the method well ablated. However, to make it a good submission, I think it would be important to learn about the following aspects:
- How sensible is the method to the threshold δc across datasets?
- How does a successful discovery of a mask influence the next iteration of the policy network?
- It would be great to also have failure cases of the method

I think the outlined points are important to be addressed before acceptance, but I like the idea and it works well. Therefore, I will give a weak accept.

---

> ### Author Response · Authors · 2024-11-26
> **Responses to Reviewer xEK4 (Part1)**
>
> We appreciate the reviewer’s thoughtful comments and address the main concerns below.
>
> ***Q1: The paper investigates the sensibility of the threshold $\delta_c$ on 1 dataset, which is fine, but it would be interesting to know if this threshold is general or needs to be tuned individually for each dataset.***
>
> **A1**: As requested, we further conduct a comprehensive ablation study on ScanNet and S3DIS to assess the sensitivity of the parameter $\delta_c$. The following Table 1 shows the results, we can see that $\delta_c$ should be typically set as 0.14 or 0.16 on two datasets. Nevertheless, since S3DIS has more occluded or distorted shapes than ScanNet, it is more challenging for our object-centric network to identify 3D objects in S3DIS. Therefore, $\delta_c$ is slightly larger (more relaxed) in S3DIS than in ScanNet.
>
> *Table1: Ablation results of different $\delta_c$ on ScanNet validation set and S3DIS Area5.*
> | Dataset        | $\delta_c$ | AP (%) | AP50 (%) | AP25 (%) |
> |----------------|------------|--------|----------|----------|
> | **ScanNet**    | 0.12       | 42.6   | 60.8     | 67.6     |
> |                | 0.14       | **46.7** | **71.5** | 82.9     |
> |                | 0.16       | 43.8   | 68.8     | 81.0     |
> |                | 0.18       | 43.8   | 70.1     | **85.0** |
> |                | 0.20       | 42.5   | 69.9     | 84.2     |
> | **S3DIS Area5**| 0.12       | 46.2   | 65.7     | 71.7     |
> |                | 0.14       | 46.4   | 66.2     | 73.8     |
> |                | 0.16       | **51.3** | **81.8** | 86.0     |
> |                | 0.18       | 48.3   | 83.1     | **90.5** |
> |                | 0.20       | 44.9   | 79.1     | 88.7     |
>
> In the revised paper, we have added the new results in Table 10 of Appendix E.
>
> ***Q2: It would be interesting to know how a successful discovery of a mask influences the next iteration of the policy network.***
>
> **A2**: We appreciate this insightful comment. To explore the influence of previously discovered masks on the training of policy networks, we analyze the quality and accuracy of discovered candidate masks from the object discovery branch (RL) at various epochs. In particular, we conduct this analysis on the training set of ScanNet. A discovered mask is considered as  accurate if it has an IoU greater than 50\% with any ground truth mask. We track the number of newly discovered masks, defined as those have never been identified in all previous epochs.
>
> The following Table 2 shows the number and accuracy of discovered masks over a certain number of training epochs. We can see that: 1) Over training, the total number of discovered masks increases, and the accuracy improves. However, the number of newly discovered masks decreases over time, suggesting that the network becomes more consistent in identifying relevant masks as training advances. 2) The accuracy of newly discovered masks declines, primarily because those objects that are easier to reconstruct can be identified in early epochs. As training progresses, the model likely attempts to discover objects that are harder to represent, which is risky and error-prone.
>
> *Table2: The number and accuracy of discovered objects over a certain number of epochs. Note that, some newly discovered masks may overlap with each other. All the numbers in this table are the counts after removing overlapping masks.*
> | No. of Epochs                  | 0    | 50   | 100  | 150  | 200  | 250  | 300  | 350  | 400  |
> |-------------------------|------|------|------|------|------|------|------|------|------|
> | **Mask Number**         | 920  | 5630 | 6981 | 7421 | 7457 | 7696 | 8133 | 7922 | 7931 |
> | **Mask Accuracy (%)**   | 16.5 | 36.2 | 41.0 | 43.5 | 42.0 | 42.2 | 44.0 | 44.7 | 45.5 |
> | **New Mask Number**     | 920  | 5257 | 4336 | 2404 | 1982 | 1716 | 1576 | 1277 | 870  |
> | **New Mask Accuracy (%)** | 16.5 | 26.7 | 15.3 | 10.2 | 7.4  | 6.0  | 5.1  | 4.3  | 4.3  |
>
> In the revised paper, we have added the new results in Table 11 of Appendix G.
>
> ***Q3: In Figure 5, it says the scene in cropped and then encoded, while in section 3.3, the authors seem to argue against random cropping. In the beginning, the container-based cropping should also be close to random right? Is the idea here that the cropping will become more targeted as the container is better navigated by the policy network? Please clear up the confusion.***
>
> **A3**: Yes, it is random cropping in the beginning, and approaches to the target in future steps guided by the policy network.
>
> In the revised paper, we have clarified this point in lines 255-256 of Section 3.3.

---

> > ### Author Response · Authors · 2024-11-26
> > **Responses to Reviewer xEK4 (Part2)**
> >
> > ***Q4: What puzzles me is how the qualitative results with diffusion prior look better in some cases that the VAE based ones (Fig 6 row 1, Fig 7 row 3), but this is not reflected in the quantitative evaluation. Could you provide an intuition for this is the case? Can you also show failure cases?***
> >
> > **A4**: This is a very interesting point. As requested, we closely investigate the nuanced difference between our two models on both ScanNet and S3DIS Area5. In particular, we calculate detailed AP scores at finer thresholds. The following Table 3 shows the results. We can see that our GOPS(Diffusion) tends to achieve higher scores at AP90 and AP80 than GOPS(VAE), but relatively lower scores at AP60 and AP50. This means that GOPS(Diffusion) is more likely to segment objects in higher quality (finer boundaries), while GOPS(VAE) tends to identify objects with coarser masks. This explains why some of the qualitative results from GOPS(Diffusion) look better visually.
> >
> > *Table3: Detailed results of our method on ScanNet validation set and S3DIS Area5.*
> > |                    |            | AP90 (%) | AP80 (%) | AP70 (%) | AP60 (%) | AP50 (%) |
> > |-----------------|------------------|----------|----------|----------|----------|----------|
> > | **ScanNet**     | GOPS(VAE)        | 16.7     | 34.2     | 48.8     | 60.1     | 71.5     |
> > |                 | GOPS(Diffusion)  | 17.8     | 34.7     | 48.2     | 62.0     | 70.6     |
> > | **S3DIS Area5** | GOPS(VAE)        | 13.7     | 32.5     | 53.4     | 62.8     | 66.2     |
> > |                 | GOPS(Diffusion)  | 14.4     | 35.5     | 49.3     | 56.5     | 58.0     |
> >
> > ***Q5: It would be nice if acronyms like SDF would be introduced. Even though this is an established method, its still also done for acronyms like ViT=Vision Transformer.***
> >
> > **A5**: Sure. All acronyms have been introduced in the revised paper.
> >
> > ***Q6: How sensible is the method to the threshold $\delta c$ across datasets?***
> >
> > **A6**: Refer to A1.
> >
> > ***Q7: How does a successful discovery of a mask influence the next iteration of the policy network?***
> >
> > **A7**: Refer to A2.
> >
> > ***Q8: It would be great to also have failure cases of the method.***
> >
> > **A8**: We have shown failure cases in Figure 10 of Appendix F in the revised paper.
> >
> > In particular, the failure cases in our method mainly include two types. The first type is that our model mistakenly segments objects whose shapes are similar to the target shape (e.g., chairs). For instance, it may incorrectly segment parts of a wall with a plane as a chair. The second type is missing some occluded chairs, primarily because those severely occluded chairs are hard to be reconstructed by the object-centric network.

---

> ### Comment · Reviewer_xEK4 · 2024-11-26
> **Response to my comments & Score update**
>
> Dear authors,
> Thank you for addressing my concerns adequately and in sufficient detail. I also saw that failure cases were added to the revised version.
> I am convinced this paper is interesting to the community and ready for acceptance, I will therefore update my score to 8 accept.

---

> > ### Author Response · Authors · 2024-11-26
> > **Thank you**
> >
> > Dear reviewer xEK4,
> >
> > We highly appreciate your time and efforts on our paper. Your valuable suggestions and insights have significantly helped us to improve our manuscript.
> >
> > Best,
> > All authors

---

### Official Review · Reviewer_MM1f · 2024-11-04

**Soundness:** 2
**Presentation:** 3
**Contribution:** 2
**Rating:** 8
**Confidence:** 4

**Summary:**

The paper presents an unsupervised method to perform object detection on 3D scans based on reinforcement learning with an object prior model trained to generate objects of a specific category. The model performs a search on the 3D scan based on a policy network trained with reinforcement learning using as a reward the reconstruction quality obtained from a pre-trained generative model. The paper presents several experiments on real and synthetic datasets.

**Strengths:**

- The paper addresses an important problem, unsupervised object detection, since in most real-world scenarios labels are not available for training.
- The paper is well-written and easy to follow.
- The use of reinforcement learning is novel and an interesting idea that might be a useful tool to solve other problems on 3D scene understanding.

**Weaknesses:**

Although I think some of the ideas presented in the paper might have value for the community, I believe the framing of the paper and the evaluation is not adequate, and important baselines are missing. In the following paragraphs, I list my main concerns in detail:

- The method is presented as an unsupervised method. However, it relies on annotated data to train the generative model. Therefore, the method is not unsupervised, but weakly supervised, and has a greater advantage over other methods such as Unscene3D. In the paper is stated that those methods have an advantage since only the annotations of the correct class are kept, but those methods are designed to detect any object in the scene while the proposed method is trained specifically to detect a single type of object.

- The reinforcement learning search of objects in the scene will stop when an object is found. To find all objects in the scene it will require several starting positions for different searches. In the paper is indicated that several searches in parallel are used during training, however, this hyperparameter is not evaluated in the paper. An ablation study of this parameter and how many initialization are need to find all objects in the scene will help the reader understand the behavior of the method better.

- The reinforcement search will not be able to find all the objects in the scene in many cases. This is solved by using the objects found as pseudo-labels to train an instance segmentation model. However, this step I believe is not used for EFEM which also suffers from missing objects in the scene. This combination should be tested to show the effectiveness of the reinforcement learning algorithm. If not, we could train an instance segmentation model on the output of EFEM.

- The proposed method trains a Mask3D model on the pseudo labels generated by the reinforcement learning algorithm. However, Mask3D relies on superpoints to perform the instance segmentation prediction. The same superpoints are the ones used to annotate the labels in ScanNet, which gives Mask3D an unfair advantage over other methods since Mask3D then uses the perfect boundaries of the objects. Since the proposed method is based on Mask3D, it also has the same unfair advantage, which might explain the big improvement on ScanNet.

- The synthetic dataset is only evaluated against EFEM and not Unscene3D or Part2Object. These baselines should be included.

- The paper fails to cite in the related work a relevant work that also used a search on the scene to perform object detection based on an object pre-trained network:

Finding your (3D) center: 3D object detection using a learned loss
D Griffiths, J Boehm, T Ritschel
European Conference on Computer Vision

**Questions:**

I would like to hear the opinion of the authors on the concerns I raised in the weaknesses section and clarify possible misunderstandings in my evaluation.

**Details Of Ethics Concerns:**

No concerns

---

> ### Author Response · Authors · 2024-11-26
> **Responses to Reviewer MM1f (Part1)**
>
> We appreciate the reviewer's valuable comments and address the concerns below.
>
> ***Q1: The method is presented as an unsupervised method. However, it relies on annotated data to train the generative model. Therefore, the method is not unsupervised, but weakly supervised, and has a greater advantage over other methods such as Unscene3D. In the paper is stated that those methods have an advantage since only the annotations of the correct class are kept, but those methods are designed to detect any object in the scene while the proposed method is trained specifically to detect a single type of object.***
>
> **A1**: Thanks for the valuable comment. To avoid confusion, we have updated our paper title to ``GOPS: Learning Generative Object Priors for 3D Instance Segmentation without Scene Supervision" in the revised version. This new title exactly describes our pipeline.
>
> In addition, in lines 360-361 of Section 4.1 in the revised paper, we have also rephrased the description about assigning class labels to baselines, particularly removing the statement ``... in favor of them ".
>
> ***Q2: The reinforcement learning search of objects in the scene will stop when an object is found. To find all objects in the scene it will require several starting positions for different searches. In the paper is indicated that several searches in parallel are used during training, however, this hyperparameter is not evaluated in the paper. An ablation study of this parameter and how many initialization are need to find all objects in the scene will help the reader understand the behavior of the method better.***
>
> **A2**:  Thanks for this suggestion. As requested, we further conduct an ablation study about the number of trajectories created in parallel for the object discovery branch. Here we choose 25/50/75/100 trajectories respectively, whereas we choose 50 in our main experiments. We use the VAE-based object-centric network.
>
> The following Table 1 shows the results. We can see that: 1) the number of trajectories is not crucial once it is more than a certain number, e.g., 50. 2) Too few trajectories can lead to a decrease in the final performance due to an insufficient number of object masks discovered.
>
> *Table 1: Ablation results on ScanNet validation set for different numbers of parallel trajectories.*
> | No. of Trajectories | AP (%) | AP50 (%) | AP25 (%) |
> |---------------------|--------|----------|----------|
> | 25                  | 42.0   | 64.1     | 74.4     |
> | 50                  | 46.7   | 71.5     | 82.9     |
> | 75                  | 46.9   | 69.5     | 80.8     |
> | 100                 | 47.1   | 69.7     | 81.3     |
>
> In the revised paper, we have added the new results in Table 8 of Appendix C.

---

> > ### Author Response · Authors · 2024-11-26
> > **Responses to Reviewer MM1f (Part2)**
> >
> > ***Q3: The reinforcement search will not be able to find all the objects in the scene in many cases. This is solved by using the objects found as pseudo-labels to train an instance segmentation model. However, this step I believe is not used for EFEM which also suffers from missing objects in the scene. This combination should be tested to show the effectiveness of the reinforcement learning algorithm. If not, we could train an instance segmentation model on the output of EFEM.***
> >
> > **A3**: This is a great suggestion. In the revised paper, we have added the following experiments.
> >
> > ****(1) Add a new baseline EFEM$_{mask3d}$****: we further build a baseline by training a Mask3D model using the discovered pseudo labels from EFEM. This model maintains the same architecture and train/test settings as our object segmentation branch.
> >
> > ****(2) Only use our Object Discovery Branch****: Regarding our pipeline, once the object discovery branch is well-trained by RL, the policy network alone can discover multiple objects on point clouds by querying the frozen object-centric network (e.g., our VAE version) in testing. Intuitively, given more trajectories during testing, the object discovery branch is likely to identify more objects. For a comparison, we directly test our well-trained object discovery branch, given 50/100/300/600 trajectories respectively, denoted as GOPS(Ours-VAE)$_{dis50/..}$. Note that, the original EFEM uses 600 trajectories on each scene when discovering objects.
> >
> > The following Table 2 (also Table 1 in the revised paper) shows the results. We can see that: 1) When training a Mask3D model using pseudo labels discovered by EFEM, the performance is clearly inferior to our method, meaning that the quality of discovered objects from our pipeline is higher than EFEM. 2) When only using our object discovery branch, the final segmentation performance is not satisfactory, potentially due to missing detections. This means that training an object segmentation branch is indeed necessary and beneficial.
> >
> > *Table 2: Quantitative results of our method and baselines on the validation set of ScanNet.*
> > | Method                          | AP (%) | AP50 (%) | AP25 (%) |
> > |---------------------------------|--------|----------|----------|
> > | EFEM$_{mask3d}$           | 38.8   | 55.1     | 63.8     |
> > | GOPS (Ours-VAE)$_{dis50}$ | 26.3   | 50.7     | 56.9     |
> > | GOPS (Ours-VAE)$_{dis100}$| 26.9   | 51.2     | 59.1     |
> > | GOPS (Ours-VAE)$_{dis300}$| 28.5   | 55.2     | 66.8     |
> > | GOPS (Ours-VAE)$_{dis600}$| 28.7   | 56.2     | 66.9     |
> > | GOPS (Ours-VAE)                 | 46.7   | **71.5** | **82.9** |
> > | **GOPS (Ours-Diffusion)**       | **47.1**| 70.6     | 81.1     |
> >
> > ***Q4: The proposed method trains a Mask3D model on the pseudo labels generated by the reinforcement learning algorithm. However, Mask3D relies on superpoints to perform the instance segmentation prediction. The same superpoints are the ones used to annotate the labels in ScanNet, which gives Mask3D an unfair advantage over other methods since Mask3D then uses the perfect boundaries of the objects. Since the proposed method is based on Mask3D, it also has the same unfair advantage, which might explain the big improvement on ScanNets.***
> >
> > **A4**: This is a helpful point. We further conduct experiments to assess the impact of superpoints provided by ScanNet dataset. In particular, we use the following two new strategies for a comparison: 1) using superpoints generated by SPG [1] in an unsupervised manner, and 2) directly extracting features on voxels instead of superpoints.
> >
> > The following Table 3 shows results on the validation set of ScanNet. We can see that directly using voxels without any superpoints can achieve comparable performance with that of ScanNet superpoints, though the latter is slightly better.
> >
> > *Table 3: Ablation results of different types of superpoints on the validation set of ScanNet.*
> > | Type of Superpoints   | AP (%) | AP50 (%) | AP25 (%) |
> > |-----------------------|--------|----------|----------|
> > | **ScanNet superpoints** | **46.7** | **71.5** | **82.9** |
> > | SPG superpoints       | 43.7   | 61.9     | 69.1     |
> > | Without superpoints   | 45.1   | 65.2     | 72.3     |
> >
> > In the revised paper, we have added the new results in Table 9 of Appendix D.

---

> ### Author Response · Authors · 2024-11-26
> **Responses to Reviewer MM1f (Part3)**
>
> ***Q5: The synthetic dataset is only evaluated against EFEM and not Unscene3D or Part2Object. These baselines should be included.***
>
> **A5**: Thanks for the advice. For Unscene3D and Part2Object, both require paired RGB images to extract 2D features via pretrained DINO/v2 for training their own detection networks, so it is unable to directly train them on our synthetic dataset due to the lack of paired RGB images.
>
> For reference, we directly reuse their models well-trained on the training set of ScanNet in Section 4.1, and then test on our synthetic dataset. Since such a setting is not strictly fair to them, we group them as the category **Unsupervised\&Real2Syn**. For a comparison, we also reuse our model (VAE version) well-trained on the *chair* category of ScanNet training set in Section 4.1, and directly test on our synthetic dataset, denoted as **GOPS(Ours-VAE)$_{real2syn}$** .
>
> The following Table 4 (also Table 5 in the revised paper) shows the results. We can see that, in the setting of **real2syn**, our  **GOPS(Ours-VAE)$_{real2syn}$**  is better than Unscene3D which uses both DINO [3] and CSC [4] features, but falls short of Part2Object which uses the sophisticated DINOv2 [2] features. This is quite understandable because our **GOPS(Ours-VAE)$_{real2syn}$** mainly leverages the limited object priors of a single *chair* class from ShapeNet, whereas Part2Object uses much richer objectness priors from DINOv2. We leave the exploration of richer 2D or 3D object priors for our future work.
>
> *Table4: Quantitative results on the test set of our synthetic dataset.*
> | Category                  | Method                        | AP (%) | AP50 (%) | AP25 (%) |
> |---------------------------|-------------------------------|--------|----------|----------|
> | **Unsupervised & Real2Syn**   | Part2Object                   | 46.1   | 69.3     | 81.5     |
> |                                               | Unscene3D                     | 37.7   | 59.7     | 76.2     |
> |                                               | **GOPS(Ours-VAE)$_{real2syn}$** | 38.8   | 59.3     | 65.8     |
> |  **Unsupervised**                                | EFEM                          | 20.7   | 34.1     | 46.6     |
> |                           | GOPS(Ours-VAE)                | 58.7   | 85.0     | 90.6     |
> |                           | GOPS(Ours-Diffusion)          | 58.5   | 85.9     | 91.5     |
>
> In the revised paper, we have updated the experiment settings in lines 426-431 and results in Table 5 of Section 4.3.
>
> ***Q6: The paper fails to cite in the related work a relevant work that also used a search on the scene to perform object detection based on an object pre-trained network:
> Finding your (3D) center: 3D object detection using a learned loss D Griffiths, J Boehm, T Ritschel European Conference on Computer Vision***
>
> **A6**: Thanks for sharing the relevant work which has been discussed in Section 2 of our revised paper.
>
> **References**
>
> [1]. Loic Landrieu and Martin Simonovsky. Large-scale point cloud semantic segmentation with superpoint graphs. CVPR, 2018.
>
> [2]. Maxime Oquab, Timoth´ee Darcet, Th´eo Moutakanni, Huy V Vo, Marc Szafraniec, Vasil Khali-
> dov, Pierre Fernandez, Daniel Haziza, Francisco Massa, Alaaeldin El-nouby, Mahmoud Assran,
> Nicolas Ballas, Wojciech Galuba, Russell Howes, Po-yao Huang, Shang-wen Li, Ishan Misra,
> Michael Rabbat, Vasu Sharma, Gabriel Synnaeve, Hu Xu, Herv´e Jegou, and Julien Mairal. DI-
> NOv2: Learning Robust Visual Features without Supervision. TMLR, 2024
>
> [3]. Mathilde Caron, Hugo Touvron, Ishan Misra, Herv´e J´egou, Julien Mairal, Piotr Bojanowski, and
> Armand Joulin. Emerging Properties in Self-Supervised Vision Transformers. ICCV, 2021
>
> [4]. Ji Hou, Benjamin Graham, Matthias Nießner, and Saining Xie. Exploring data-efficient 3d scene
> understanding with contrastive scene contexts. CVPR, 2021.

---

> > ### Comment · Reviewer_MM1f · 2024-11-27
> > **Great New Experiments**
> >
> > First, I thank the authors for the extensive work they invested in including all my points in the revised version of the paper. I believe the additional experiments provided by the authors greatly improve the paper and remove any possible concerns about the improvements provided by this method. Therefore, I will raise my score and hope the paper can be presented at the conference.

---

> > > ### Author Response · Authors · 2024-11-27
> > > **Thank you**
> > >
> > > Dear reviewer MM1f,
> > >
> > > Thank you very much for your time and efforts on our paper. Your suggested baselines, experiments and all other insightful comments really help us improve our manuscript.
> > >
> > > Best, All authors

---

### Meta-Review · Area_Chair_9PKB · 2024-12-18

**Metareview:**

The paper proposes a pipeline with several modules to find objects in 3D scenes without any human labels, which is important because many real-world situations lack labeled data for training. A policy network trained with reinforcement learning is used as a reward for the reconstruction quality obtained from a pre-trained generative model to search the entire 3D point cloud for objects. Reviewers all agree that the paper is well-motivated and clearly presented. Detailed ablations have been conducted to analyze different aspects of the pipeline. All reviews are positive, with ratings of 8, 8, and 6. The area chair concurs with the reviewers' decisions and, therefore, recommends accepting the paper.

**Additional Comments On Reviewer Discussion:**

The authors addressed the reviewers' concerns by providing additional results, which include
* Various quantitative results and class-agnostic segmentation results on the validation set of ScanNet and S3DIS Area5;
* Quantitative results on the test set of our synthetic dataset;
* Ablation results of different types of superpoints,  different threshold values, varying numbers of parallel trajectories, and attention blocks in the object-centric network;
* Analysis of the number and accuracy of discovered objects over a certain number of epochs;
* Evaluation of the object-centric network for shape reconstruction.

---

### Decision · Program_Chairs · 2025-01-22

Accept (Spotlight)